# SLC4A1 mutations that cause distal renal tubular acidosis alter cytoplasmic pH and cellular autophagy

Grace Essuman[1], Midhat Rizvi[1], Ensaf Almomani[2], Shahid AKM Ullah[1,3], Sarder MA Hasib[4], Forough Chelangarimiyandoab[1], Priyanka Mungara[1], Manfred J Schmitt[4], Marguerite Hureaux[5], Rosa Vargas-Poussou[5], Nicolas Touret[6], Emmanuelle Cordat[1]*

[1]Department of Physiology, University of Alberta, Edmonton, Canada; [2]Department of Basic Medical Sciences, Faculty of Medicine, Al-Balqa Applied University, Al-Salt, Jordan; [3]Department of Medicine, University of Alberta, Edmonton, Canada; [4]Department of Molecular and Cell Biology, Department of Biosciences (FR 8.3) and Center of Human and Molecular Biology (ZHMB), Saarland University, Saarbrücken, Germany; [5]Department of Genetics, Georges Pompidou European Hospital, Paris, France; [6]Department of Biochemistry, University of Alberta, Edmonton, Canada

*For correspondence:
cordat@ualberta.ca

## eLife Assessment

This work reports the characterization of newly identified genetic variants of SLC4A1 in patients with distal renal tubular acidosis. Cell culture studies supplemented with histological analysis of a previously established disease mouse model provide **convincing** evidence that some of the variants increase intracellular pH, reduce ATP synthesis, and attenuate autophagic degradative flux. The study is **valuable** in establishing a mechanistic framework for future exploration of the link between intracellular pH and mutations in SLC4A1 in vivo.

**Abstract** Distal renal tubular acidosis (dRTA) is a disorder characterized by the inability of the collecting duct system to secrete acids during metabolic acidosis. The pathophysiology of dominant or recessive *SLC4A1* variant-related dRTA has been linked with the mis-trafficking defect of mutant kAE1 protein. However, in vivo studies in kAE1 R607H dRTA mice and humans have revealed a complex pathophysiology implicating a loss of kAE1-expressing intercalated cells and intracellular relocation of the $H^+$-ATPase in the remaining type-A intercalated cells. These cells also displayed accumulation of ubiquitin and p62 autophagy markers. The highly active transport properties of collecting duct cells require the maintenance of cellular energy and homeostasis, a process dependent on intracellular pH. Therefore, we hypothesized that the expression of dRTA variants affects intracellular pH and autophagy pathways. In this study, we report the characterization of newly identified dRTA variants and provide evidence of abnormal autophagy and degradative pathways in mouse inner medullary collecting duct cells and kidneys from mice expressing kAE1 R607H dRTA mutant protein. We show that reduced transport activity of the kAE1 variants correlated with increased cytosolic pH, reduced ATP synthesis, attenuated downstream autophagic pathways pertaining to the fusion of autophagosomes and lysosomes and/or lysosomal degradative activity. Our study elucidated a close relationship between the expression of defective kAE1 proteins, reduced mitochondrial activity, and decreased autophagy and protein degradative flux.

## Introduction

Distal renal tubular acidosis (dRTA) is a disorder characterized by the inability of the collecting duct system to secrete acids during metabolic acidosis (*Giglio et al., 2021*). In addition to hyperchloremic metabolic acidosis, patients with this disease can present with hypokalemia, kidney stones, urinary sodium waste, and difficulty thriving. Expression of pathogenic variants in the *ATP6V0A4*, *ATP6V1B1*, *FOXI1*, *WDR72*, and *SLC4A1* genes are the usual genetic etiologies (*Escobar et al., 2016*; *Rungroj et al., 2018*; *Enerbäck et al., 2018*). The *SLC4A1* gene encodes the anion exchanger 1 (AE1) protein, which is an electroneutral chloride/bicarbonate exchanger (*Toye et al., 2004*). It exists in two forms: a 911 amino acid erythroid isoform known to interact with erythroid cytoskeletal proteins and participate in red cell respiration and integrity, and a 65 amino acid ($NH_2$-terminal) truncated isoform primarily found in the basolateral membrane of renal type A intercalated cells (A-IC) (*Kollert-Jons et al., 1993*; *Nuiplot et al., 2015*) and podocytes (*Wu et al., 2010*). This isoform participates in bicarbonate reabsorption and through its physical and functional interaction with the cytosolic carbonic anhydrase II and apical $H^+$-ATPase, it supports apical proton export and urine acidification (*Cordat and Reithmeier, 2014*).

Pathogenic variants in the *SLC4A1* gene can result in either red cell defects (such as Southeast Asian ovalocytosis [*Sawasdee et al., 2006*; *Cheung et al., 2005*] and hereditary spherocytosis [*Tang et al., 2020*]), renal cell defects (dRTA; *Sawasdee et al., 2006*) or both in patients with homozygous (Band 3 Coimbra and Band 3 Courcouronnes; *Ribeiro et al., 2000*; *Toye et al., 2008*) or compound heterozygous variants (*Chang et al., 2009*; *Khositseth et al., 2012*). Renal *SLC4A1* disease-causing variants have only been found in the transmembrane domain—where it could impact protein structure and its transport function—or in the short carboxyl domain—where it possibly affects protein–protein interactions. The pathophysiology of dominant or recessive *SLC4A1*-related dRTA (hereafter named dRTA) has originally been linked with the mis-trafficking defect of mutant kAE1 protein (*Sawasdee et al., 2006*; *Cordat, 2006*). However, recent in vivo studies in R607H (orthologous to human R589H dRTA variant) and L919X knockin mice and dRTA patients have revealed a complex pathophysiology where kAE1-expressing intercalated cells were lost, and in the remaining type-A intercalated cells, the $H^+$-ATPase relocated intracellularly and accumulated autophagy marker p62 and ubiquitin-positive material (*Mumtaz et al., 2017*).

The highly active transport properties of collecting duct cells require the tight maintenance of cellular energy and homeostasis. The autophagy-mediated turnover of damaged organelles is necessary for protecting collecting duct cells as in most renal cells (*Festa et al., 2018*). The chloride/bicarbonate exchange function of kAE1 in A-ICs confers a pivotal role in pH homeostasis and thus is a major contributor to cellular homeostasis. kAE1 protein interacts with several proteins such as integrin-like kinase (ILK), adaptor-related protein complex 1, 3, and 4 (AP-1, AP-3, and AP-4 mu1A), transmembrane protein 139 (TMEM139), kinesin family member 3B (KIF3B), and clathrin, among others (*Nuiplot et al., 2015*; *Keskanokwong et al., 2007*; *Duangtum et al., 2011*; *Sawasdee et al., 2010*) that support protein stability and trafficking. It also interacts with homeostatic proteins including the glycolytic enzyme glyceraldehyde-3-phosphate dehydrogenase (GAPDH) *Su et al., 2011* and the antioxidant enzyme peroxiredoxin 6 (PRDX 6) (*Sorrell et al., 2016*), which play major roles in cellular energy metabolism and oxidative stress response, respectively.

In this study, we report the characterization of newly identified dRTA genetic variations and provide evidence of abnormal autophagy and degradative pathways in cells and kidneys from mice expressing dRTA mutant kAE1 proteins.

## Results

### The dRTA kAE1 variants traffic to the basolateral membrane but have reduced transport activity in mIMCD3 cells

We first characterized three newly identified dRTA mutations and compared them with kAE1 WT or previously characterized kAE1 R589H mutant. *Figure 1A* depicts the alpha fold predicted structure of kAE1 showing amino acids mutated in each kAE1 mutant. kAE1 R295H is a recessively inherited substitution in the N-terminal cytosolic domain of the protein. kAE1 Y413H is a dominantly inherited substitution in transmembrane domain (TM) 1 of the core domain. In the gate domain, dominantly inherited S525F and R589H substitutions occur in TM 5 and TM 6, respectively. We generated the

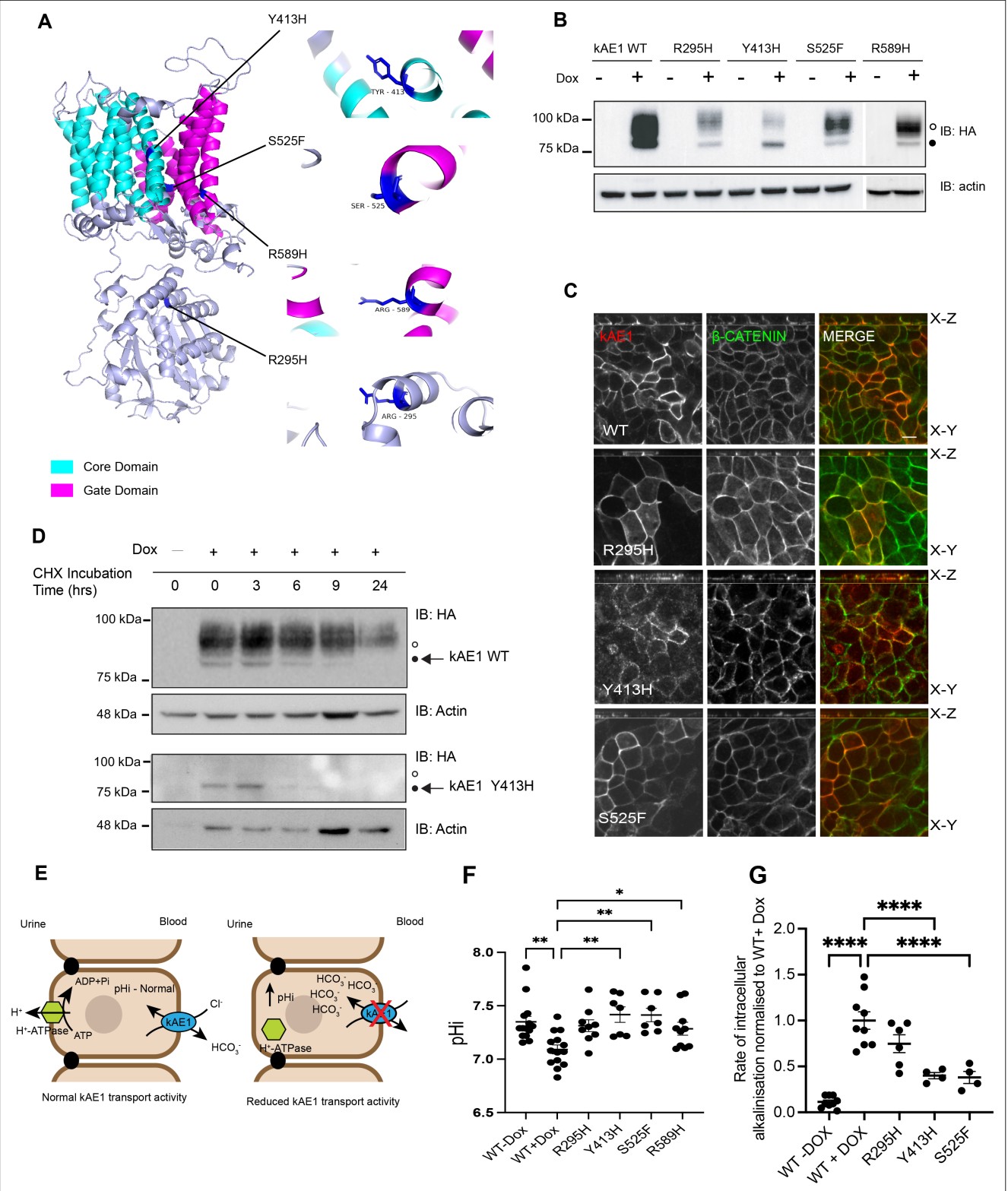

**Figure 1.** kAE1 R295H, Y413H, S525F, and R589H dRTA mutants are either dysfunctional or prematurely degraded. (**A**) Alpha Fold predicted structure of the kidney isoform of Band 3 anion exchanger 1 (kAE1) with core and gate domains highlighted. The dRTA kAE1 mutation sites are colored in blue with line extensions detailing specific amino acids mutated. (**B**) Immunoblot showing expression of kAE1 WT, R295H, Y413H, S525F, and R589H and corresponding actin band in mIMCD3 cells treated with and without doxycycline for 24 hours. Mouse anti-HA antibody was used to detect kAE1-HA, top (open circle) and bottom bands (closed circle) correspond to kAE1 carrying complex and high mannose oligosaccharides, respectively.

*Figure 1 continued on next page*

*Figure 1 continued*

(**C**) Immunostaining of kAE1 WT or mutant (red) and β-catenin (green) in polarized mIMCD3 cells. Scale bar = 10 μm. Red = kAE1, green = ß-catenin. (**D**) Immunoblot of cycloheximide (CHX) chase assay with corresponding actin in kAE1 mIMCD3 WT and Y413H cells showing the degradation of kAE1 Y413H after 3 hours CHX incubation. (**E**) Cartoon depicting the transporter activity and expected changes in pHi in cells expressing kAE1 WT (left) or inactive mutant (right). (**F**) Graphical representation of intracellular pH (pHi) measurement of mIMCD3 kAE1 WT, R295H, Y413H, S525F, and R589H cells. Error bars correspond to mean ± SEM, n=minimum 32. *p<0.05, **p<0.01 using one-way ANOVA followed by a Dunnett's post hoc test. (**G**) Rate of intracellular alkalinization in WT or mutant mIMCD3 cells normalized to WT + Dox. **** indicates p<0.0001 using one-way ANOVA followed by a Dunnett's post hoc test. Error bars correspond to mean ± SEM, n=minimum 4.

newly identified dRTA variant cDNAs and expressed them or kAE1 WT in mIMCD3 cells. As seen on *Figure 1B*, kAE1 R295H, S525F, and R589H variants display two typical bands similar to kAE1 WT. The top band (open circle) encompasses kAE1 proteins carrying a complex oligosaccharide that have reached the Golgi and beyond, while the bottom band (black circle) corresponds to high mannose-carrying kAE1 proteins located in the endoplasmic reticulum. However, kAE1 Y413H mutant bands intensity was overall weaker than WT and displayed a predominant single band aligned with high mannose-carrying kAE1 proteins. These results indicate that the three newly described dRTA mutants are successfully expressed in mIMCD3 cells. We next localized these mutants by immunofluorescence in polarized mIMCD3 cells. Both kAE1 WT and mutants appropriately co-localized with basolateral membrane marker beta-catenin in polarized mIMCD3 cells (*Figure 1C*). kAE1 R589H location has previously been reported at the basolateral membrane in polarized mIMCD3 cells (*Cordat, 2006*). However, staining for kAE1 Y413H was again weaker and seemed more intracellular than other mutants. To address a possible premature degradation of this variant, we measured its lifetime and observed that its degradation begins 6 hours post-synthesis while kAE1 WT abundance remained stable for 24 hours (*Figure 1D*). Finally, to assess the transport activity of the mutants, we examined the steady-state cytosolic pH (pHi) and rate of intracellular alkalinization of mIMCD3 cells expressing kAE1 WT or mutants (*Figure 1E–G*). Using BCECF-AM, we observed that the steady state intracellular pH (pHi) of kAE1 mutant cells was more alkaline than WT cells (except for kAE1 R295H cell pHi, which has a similar trend but is not significantly different from WT) (*Figure 1F*), in agreement with rate of intracellular alkalinization (*Figure 1G*). Note that transport activity, measured as rate of alkalinization, is measured by reverting kAE1 exchange activity from bicarbonate *export* to *import* (see 'Materials and methods, protocol for transport assay), hence a reduced alkalinization rate is observed in cells expressing defective kAE1 protein compared to WT (*Figure 1G*). Overall, these results indicate that except for the kAE1 Y413H mutant, the other newly described variants are expressed and traffic to the basolateral membrane of polarized mIMCD3 cells, similar to the previously published kAE1 R589H mutant. Given that the premature degradation of kAE1 Y413H mutant likely explains the dRTA phenotype, we did not perform further assays on cells expressing this protein.

## Autophagy processes are altered in mIMC3 cells expressing the kAE1 R295H, S525F, and R589H dRTA mutants and in R607H knock-in kidney lysates

In mice expressing kAE1 R607H (the murine equivalent to human dRTA R589H substitution), a striking reduction in type-A intercalated cells was noted, and in the remaining cells, autophagy marker p62 and ubiquitin accumulated in these abnormally enlarged cells (*Mumtaz et al., 2017*). We therefore investigated the autophagy machinery in dRTA mutant mIMCD3 cells and in R607H knock-in (KI) mice. We first examined the ratio of autophagosome marker LC3BII protein relative to the total intensities of LC3BI and LC3BII as well as p62 levels. These experiments were performed at steady state, upon autophagy induction by starvation, or after autophagy inhibition by Bafilomycin (Baf) A1 (*Figure 2A–C*). *Figure 2A–I* shows a consistent increase in the ratio of LC3B II to total LC3B (I+II) in the mutant cells at steady state (*Figure 2D*), with starvation (*Figure 2F*) and with Baf A1 (*Figure 2H*) except for the kAE1 R295H mutant which was not significantly different from WT. This suggests an altered autophagy process in the mutant cells, in agreement with preliminary findings from Mumtaz and colleagues (*Mumtaz et al., 2017*). There was no significant difference in p62 abundance in mutant cells compared to WT at steady state and with starvation (*Figure 2E and G*). However, with Baf A1, R589H mutant cells had significantly lower p62 abundance compared to WT (*Figure 2I*). To confirm these findings, we next assessed the abundance of these markers in whole kidney lysates from the kAE1 R607H KI

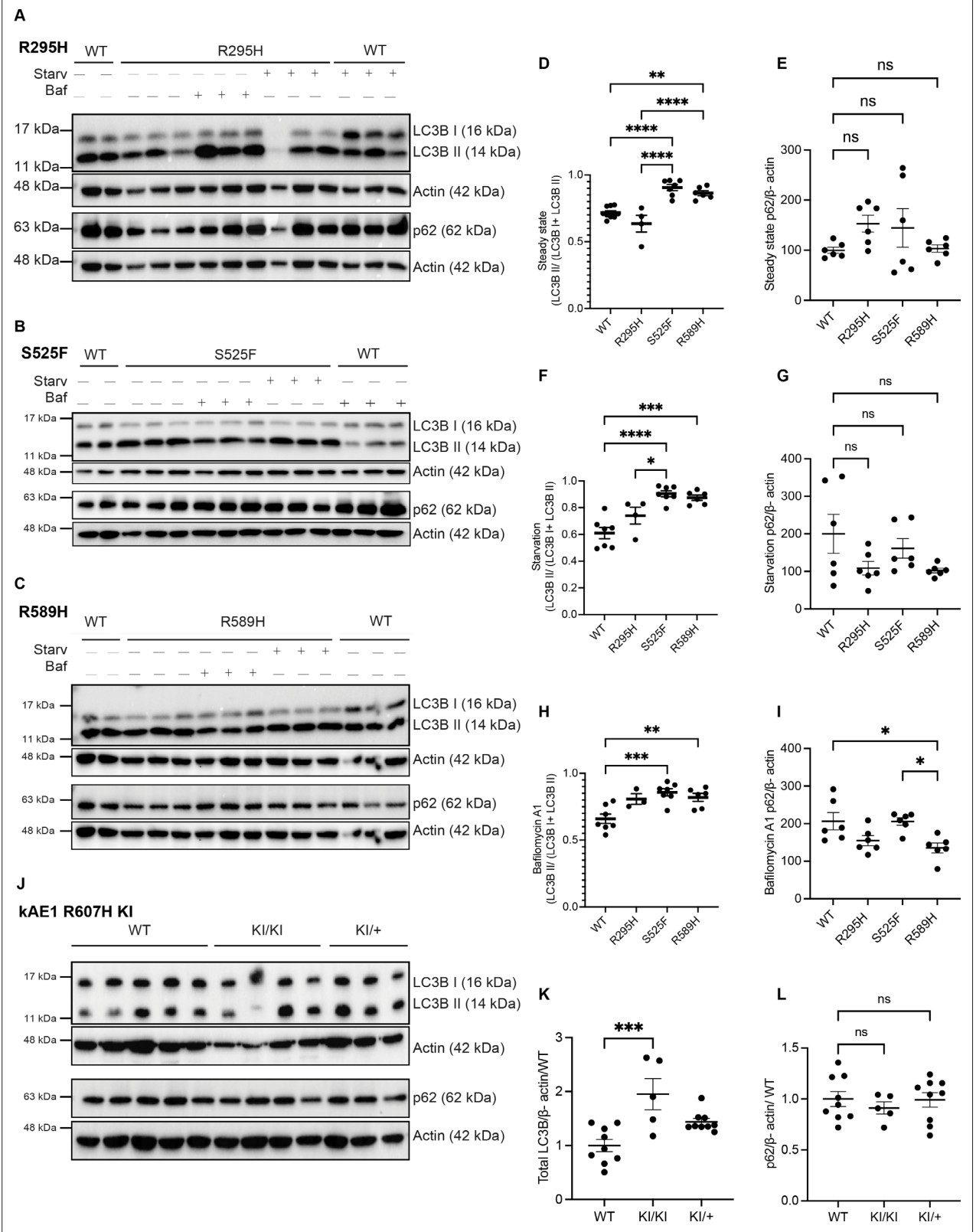

**Figure 2.** Autophagy is upregulated in dRTA kAE1 mutants in vitro and in vivo. . (**A–C**) Representative immunoblots of LC3B and p62 with corresponding actin abundance in kAE1 WT, R295H, S525F, and R589H mIMCD3 cells at steady state, under starvation (Starv) or 400 nM Bafilomycin A1 (Baf) treatment. Note that p62 and LC3B were detected on the same blot for (**A**) and (**C**); therefore, the same actin blot is shown for both panels. (**D–I**) Quantification of all immunoblots showing the ratio of LC3B II to total LC3B and p62. Error bars correspond to mean ± SEM, n=3–8. *p<0.05, ** p<0.01, ***p<0.005,

*Figure 2 continued on next page*

Figure 2 continued

****p<0.001 using one-way ANOVA followed by a Tukey's post hoc test. Immunoblots (**J**) and quantification (**K, L**) of LC3B and p62 abundance in kAE1 R607H KI mouse whole kidney lysates. Error bars correspond to mean ± SEM, n=minimum 5. ***p<0.005 using one-way ANOVA followed by a Tukey's post hoc test.

mice. We observed that the total LC3B abundance was significantly higher in homozygous KI mice (KI/KI) compared to WT, with no difference in p62 (*Figure 2J–L*). Overall, these results suggest an abnormal autophagy in mutant mIMCD3 cells and in kidneys of R607H KI mice. Given the lack of difference in phenotype between the recessive kAE1 R295H and kAE1 WT mIMCD3 cells, we focused the subsequent experiments on dominant kAE1 S525F and R589H mutant cells. Further investigations will be needed to understand the pathophysiology associated with the kAE1 R295H novel variant.

## Late autophagy steps are blocked in dRTA kAE1 mutant-expressing cells due to their alkaline intracellular pH

Given these preliminary findings of abnormal autophagy in dRTA mutant cells, we examined in more detail their autophagy machinery by transiently transfecting them with the eGFP-RFP-LC3 construct and monitoring the rate of autophagosome and autolysosome formation. In this assay, the green fluorescent protein (eGFP) fused to LC3 is quenched in the acidic environment of the autolysosome, while both eGFP and red fluorescent protein (RFP) fluoresce in vesicles in the neutral lumen of the autophagosome (*Zhou et al., 2012*; *Kimura et al., 2007*). To avoid the poor efficiency of transient transfection in polarized cells, experiments were conducted in sub-confluent cells, pending that the mutant proteins are present at the plasma membrane. We therefore performed cell surface biotinylations on 70–80% sub-confluent cells, which confirmed a robust plasma membrane abundance of both kAE1 R589H and S525F that was not significantly different from the WT protein (*Figure 3A and B*). Therefore, we next assessed the efficiency of the autophagy machinery in WT or dRTA mutant-expressing cells (*Figure 3C–F*). Focusing on cells expressing either kAE1 WT, kAE1 S525F, or R589H, we quantified the number RFP+ (red, acidic autolysosomes) and double eGFP+/RFP+ (yellow, not acidic autophagosomes) vesicles. kAE1 S525F mutant cells have significantly more autophagosomes than WT counterparts (*Figure 3D*)**,** and both kAE1 S525F and R589H have significantly more autolysosomes than WT (*Figure 3E*). *Figure 3F* shows that both mutants had significantly more autophagosomes (yellow) than autolysosomes (red). This finding suggests an upregulation of autophagy and inhibition in the downstream steps of the process that involves the fusion of autophagosomes with the lysosome (*Mizushima, 2018*). As autolysosomes require luminal v-H$^+$-ATPase-dependent acidification to efficiently clear cell debris (*Hu et al., 2022*), the higher intracellular pH seen in mutant cells (*Figure 1F*) may be detrimental to v-H$^+$-ATPase full activity and impair proper autolysosomal acidification (*Berezhnov et al., 2016*). We therefore wondered whether chemically acidifying the pHi in mutant cells would rescue the autophagy machinery (*Figure 3G–K*). We first determined that incubation of mIMCD3 cells in 0.033 μM nigericin in cell culture medium at pH 6.6 for 2 hours acidified cytosolic pH to 6.9 without causing cell death (*Figure 3—figure supplement 1*). Next, we observed that chemically reducing pHi to 6.9 in mutant expressing cells reduced the ratio of LC3B II to total LC3B (*Figure 3J*) and the abundance of lysosomal-associated membrane protein 1 (LAMP1) in R589H cells (*Figure 3K*) to levels similar to WT cells at steady state. These findings suggest that abnormal autophagy in the mutant cells may be caused by their alkaline pHi, resulting from a reduced anion exchange activity of the mutant kAE1 protein.

## mIMCD3 cells expressing dRTA kAE1 mutants and R607H KI kidney tissues have abnormal lysosome number and size

The accumulation of autophagosomes and autolysosomes as seen above may suggest one or a combination of the following: an inability of autophagosomes to fuse with lysosomes and/or a defect in lysosomal degradative activity in the mutant cells (*Button et al., 2017*; *Festa et al., 2018*). We first examined the lysosomal degradative activity by assessing lysosomal protease Cathepsin B activity using Magic Red staining, a probe that fluoresces upon lysosomal protease cleavage (*Figure 4A and B*). In agreement with increased RFP+ vesicles (*Figure 3E*), the kAE1 S525F mutant cells had a significantly higher number of Magic Red-positive vesicles than WT, whereas the kAE1 R589H mutant

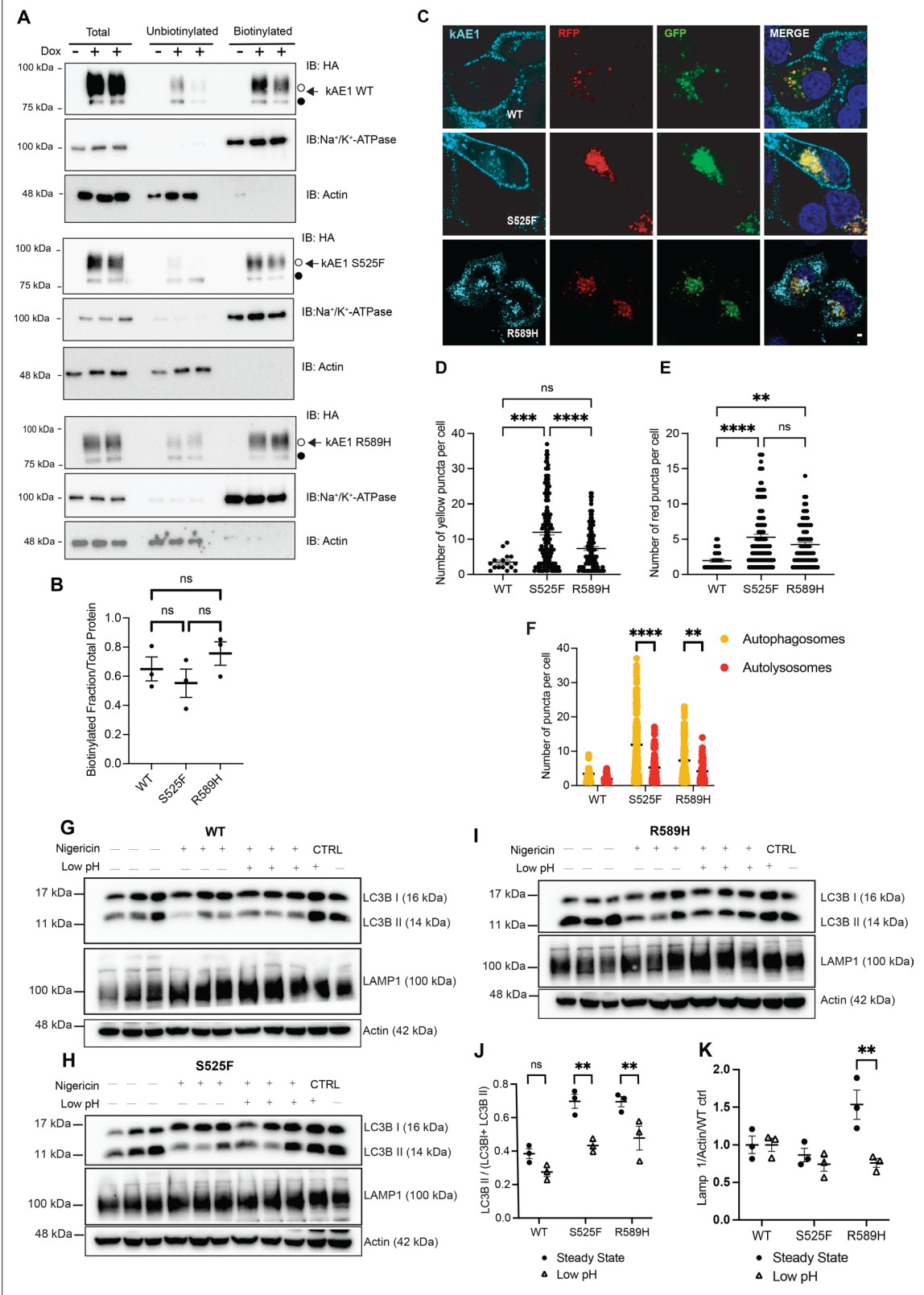

**Figure 3.** dRTA kAE1 mutants have more alkaline steady-state intracellular pH and altered autophagy flux. (**A**) Representative immunoblots of cell surface biotinylation experiments in mIMCD3 cells expressing kAE1 WT or mutants (top panels), with control staining of plasma membrane marker Na$^+$/K$^+$-ATPase (middle panel) and intracellular marker actin (bottom panel). (**B**) Quantification of three independent cell surface biotinylation experiments. Data are represented by a single representative blot for each variant. n.s., not significant using one-way ANOVA. Error bars correspond to mean ±

*Figure 3 continued on next page*

*Figure 3 continued*

SEM, n=3. (**C**) Immunofluorescence staining in eGFP-RFP-LC3 transfected mIMCD3 cells expressing kAE1. GFP = green, RFP = red, kAE1=cyan, nuclei = dark blue (merge only). Scale bar = 2 μm. Graphical representation of number of yellow (autophagosomes) (**D**) and red (autolysosomes) (**E**) puncta per cell expressing kAE1. Error bars correspond to mean ± SEM, n=minimum 32. **p<0.01, *** p<0.005, ****p<0.001 using one-way ANOVA followed by a Tukey's post hoc test. (**F**) Grouped graph of the number of yellow (autophagosomes) and red (autolysosomes) puncta per cell expressing kAE1, respectively. Note that the statistical analysis displayed only compared yellow and red groups for simplification. Error bars correspond to mean ± SEM, n=minimum 32. **p<0.01, ****p<0.001 using two-way ANOVA followed by a Sidak's post hoc test. (**G–I**) Immunoblot of LC3B, LAMP1, and actin in kAE1 WT, S525F, and R589H mIMCD3 cells at steady state and under chemically reduced intracellular pH conditions. Graphical representation of the ratio of LC3B II to total LC3B ratio (**J**) or LAMP1 (**K**) at steady state versus at low pHi in mIMCD3 kAE1 WT, S525F, and R589H. Black circles indicate steady state cells and triangles indicate low pHi cells. Error bars correspond to mean ± SEM, n=3. ** indicates p<0.01 using two-way ANOVA followed by a Sidak's post hoc test.

The online version of this article includes the following figure supplement(s) for figure 3:

**Figure supplement 1.** Two-hour incubation in pH 6.6 media with 0.033 μM nigericin reduces cytosolic pH to 6.9.

cells had significantly larger Magic Red-positive vesicles, suggesting an accumulation of undigested material (*de Araujo et al., 2020*). To validate this finding in vivo, we performed immunostaining and quantified LAMP1-positive staining in ß1 ATPase-positive cells (a marker of ICs) in WT and R607H KI mouse kidney sections. The KI mice showed significantly more and larger LAMP1-positive vesicles compared to WT mice in both cortex and medulla (*Figure 4C–F*). We probed further into the lysosomal activity by quantifying lysosomal protease Cathepsin D (immature, intermediate, and mature) protein abundance by immunoblot in isolated primary murine A-ICs. Although the abundance of immature and intermediate cathepsin D did not differ between genotypes, the KI mice showed a significantly decreased abundance of mature cathepsin D (*Figure 4—figure supplement 1*). Thus, in line with in vitro findings, A-IC from homozygous R607H KI mice display relatively more and larger lysosomes with reduced active protease abundance than WT littermates, suggesting a lysosomal defect in the dRTA kAE1 mutant cells.

## dRTA kAE1 mutant cells have lower ATP production rate and abnormal mitochondrial content

Lysosomal degradation is highly dependent on a low luminal pH generated in part by the vacuolar-type $H^+$-ATPase (*Ratto et al., 2022*) whose activity depends on ATP hydrolysis. We therefore analyzed the ATP production rate in mIMCD3 cells, specifically glycolysis and oxidative phosphorylation. We measured the oxygen consumption rate (OCR) (*Figure 5A*) and extracellular acidification rate (ECAR) (*Figure 5B*) in empty vector-transfected, kAE1 WT, or mutant cells. Both kAE1 S525F and R589H mutant cells had a lower ATP production rate compared to WT (*Figure 5C*). More specifically, the R589H mutant cells had a lower mitochondrial ATP production rate (*Figure 5D*), whereas the kAE1 S525F mutant cells exhibited a lower glycolytic ATP production rate (*Figure 5E*). With the mitochondria being the major ATP-producing organelles (*Jonckheere et al., 2012*), we assessed mitochondrial content by immunostaining of translocase of the outer membrane 20 (TOM20) both in vitro and in vivo. Both kAE1 S525F and R589H mutant cells have higher mitochondrial content compared to WT as determined by total overall intensity of TOM20-positive puncta (*Figure 5F and G*). In line with this result, although a decreased fluorescence intensity was observed in the cortex, there was a significantly higher TOM20 fluorescence intensity in medullary kidneys of homozygous R607H KI mutant mice compared to WT littermates (*Figure 5H–K*).

## Discussion

In this study, we characterized three newly identified dRTA-causing kAE1 variations. Combining in vivo and in vitro studies, we demonstrated that reduced transport activity of the kAE1 mutants correlated with increased cytosolic pH, reduced ATP synthesis, attenuated downstream autophagic pathways, and lysosomal dysfunction, pertaining to the fusion of autophagosomes and lysosomes and/or lysosomal degradative activity (see summary in *Tables 1 and 2*).

In line with previous observations in mIMCD3 cells (*Mumtaz et al., 2017*), the kAE1 R295H, Y413H, and S525F mutants were properly localized to the basolateral membrane after polarization. When expressed in Madin-Darby canine kidney (MDCK) cells, other dRTA-causing kAE1 mutants such as

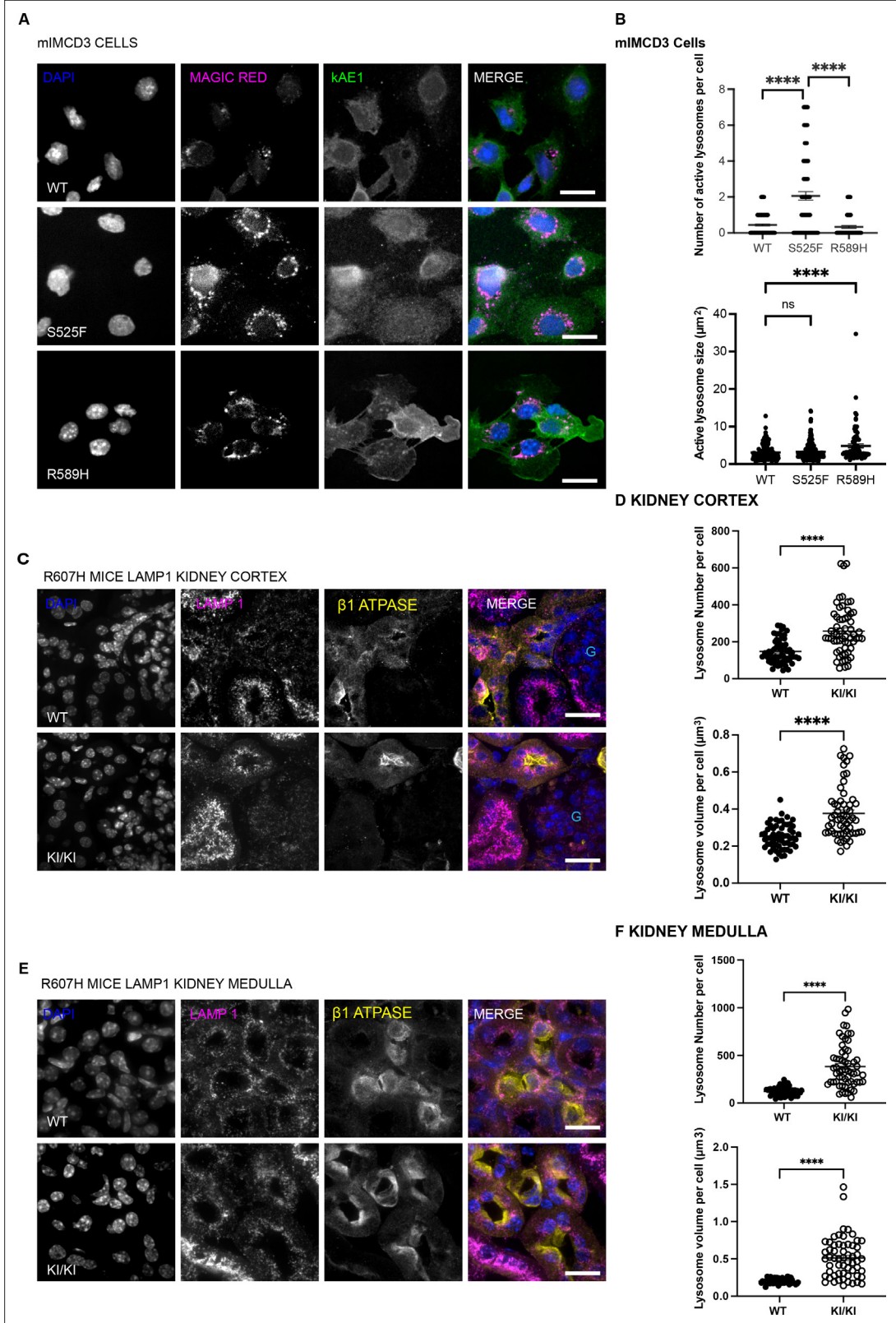

**Figure 4.** dRTA kAE1 mutants have bigger or more lysosomes than WT in vitro and in vivo. (**A**) Immunofluorescence images of kAE1 WT, S525F, and R589H mIMCD3 cells incubated with Magic Red substrate for 1 hour at 37°C in the dark. Green = kAE1, magenta = active lysosomes, blue = nuclei. Scale bar = 16 µm. (**B**) Graphical representation of number and size of active lysosomes per cell. Error bars correspond to mean ± SEM, n=minimum 30. ****p<0.0001 using one-way ANOVA followed by Tukey's post hoc test. Immunofluorescence images of LAMP1 and ß1 ATPase in kidney cortex (**C**) or

*Figure 4 continued on next page*

*Figure 4 continued*

medulla (**E**) from kAE1 WT and R607H KI mice. Blue = nuclei, magenta = LAMP1 (lysosomes), yellow = ß1 ATPase, light blue 'G' indicates the location of a glomerulus. Scale bar = 8 μm. Graphical representation of the number and volume of LAMP1 vesicles in ß1 ATPase-positive cells in the kidney cortex (**D**) or medulla (**F**) of WT or R607H KI mice. Error bars correspond to mean ± SEM, n=60. ****p<0.001 using Student's *t*-test.

The online version of this article includes the following figure supplement(s) for figure 4:

**Figure supplement 1.** dRTA kAE1 R607H knockin mice have lower abundance of intercalated cell marker B1 H+-ATPase, lysosomal protease Cathepsin D and antioxidant protein PRDX 6 in intercalated cells-enriched primary cells.

dRTA R602H, G701D, V488M, deltaV850 variants exhibited a plasma membrane trafficking defect (*Sawasdee et al., 2006*; *Yang et al., 2023*; *Deejai et al., 2022*). In contrast, the kAE1 R589H mutant was correctly targeted to the plasma membrane (*Mumtaz et al., 2017*). Functionally, cells expressing the Y413H and S525F mutants exhibit about 60% reduction in chloride/bicarbonate exchange activity compared to kAE1 WT, similar to the previously characterized recessive G701D mutant but unlike the R295H mutant (*Chu et al., 2013*). Therefore, the mechanism causing dRTA remains unclear in the case of the newly identified recessive R295H variant. Additionally, the kAE1 Y413H variant exhibited a shorter half-life than the WT counterpart, likely explaining dRTA, and thus was not further investigated. These findings add to the growing list of *SLC4A1* gene variations causing dRTA.

We next investigated the roots of the altered autophagy briefly reported in R607H KI mice (*Mumtaz et al., 2017*) using mIMCD3 cells and whole kidney lysates. In the kidneys of the KI mice, a decrease in the number of A-IC, accumulation of p62 and ubiquitinylated proteins, and enlarged remaining A-ICs suggested altered autophagy processes in dRTA mutant cells and in homozygous R607H KI mice (*Mumtaz et al., 2017*; *Chu et al., 2013*). During the autophagy process, LC3B I (a marker for autophagosomes) is converted to lipidated LC3B II (*Dhingra et al., 2018*) and p62 aggregates to facilitate the degradation of ubiquitinated proteins within the autophagosome complex (*Huang et al., 2023*). In mIMCD3 cells expressing dRTA kAE1 S525F and R589H variants, LC3B lipidation was increased compared to WT, an increase that persisted with both Baf A1 treatment and starvation. Although opposite effects were expected under inhibition or induction of autophagy, such similar effect has been previously described. In the proximal tubule of obese mice, LC3B accumulation indicating a stagnated autophagy flux was observed with both chloroquine treatment and 24-hour starvation (*Yamamoto et al., 2017*). Similarly, in NRK-52E cells, a disruption of the autophagy machinery in high cadmium-stressed cells resulted in LC3B II accumulation under either Baf A1 or rapamycin (an autophagy inducer) treatment (*Lee et al., 2017*). Although LC3B II is elevated during both increased autophagy flux and disrupted autophagy, we did not observe significant accumulation of p62 in mIMCD3 cells expressing dRTA mutants. p62 is specifically a marker of autophagy-mediated protein clearance (*Brown et al., 2021*; *Mizushima et al., 2002*; *Klionsky et al., 2012*). Therefore, p62 accumulation in R607H KI mouse kidney sections strongly suggests a compromised autophagy-mediated clearance, while the increased LC3B lipidation without significant changes in p62 in mIMCD3 cells points towards either an increased autophagic flux and/or a disrupted autophagy.

To obtain a clearer picture of the precise autophagic pathway altered in the dRTA mutants, we probed further into the different stages of autophagy and autophagy flux. We noted an accumulation of autophagosomes and autolysosomes in the S525F and R589H mutant cells. This was recapitulated in the R607H KI mice which showed significantly more and larger LAMP1-positive vesicles in both kidney cortex and medulla, suggesting a blockage in late steps of autophagy flux in both dRTA mutant cells and KI mice. Such blockage has been implicated in the pathophysiology of several diseases. In lysosomal storage disease, lysosome accumulation in proximal tubule cells is a key component in the pathways mediating epithelial dysfunction (*Festa et al., 2018*). In this study, restoring autophagy flux attenuated disease progression. Another study in SK-N-SH, RT4-D6P2T, and HeLa cells implicated autophagosome and lysosome accumulation in cellular toxicity associated with neurodegenerative diseases (*Button et al., 2017*). In agreement with altered lysosomal function, we also noted a greater abundance and size of active cathepsin B lysosomal protease vesicles in mIMCD3 cells. Increased cathepsin B activity affects lysosomal biogenesis, autophagy initiation, and cellular homeostasis (*Liu et al., 2024*). In the renal context, cathepsin B knockout mice demonstrated a higher resistance and quicker recovery from glomerular damage (*Höhne et al., 2018*), suggesting that cathepsin B accumulation may be detrimental to the cells. Increased cathepsin B abundance in the dRTA mutant cells also correlates with the accumulation of lysosomes. Overall, these findings suggest that the pathogenesis

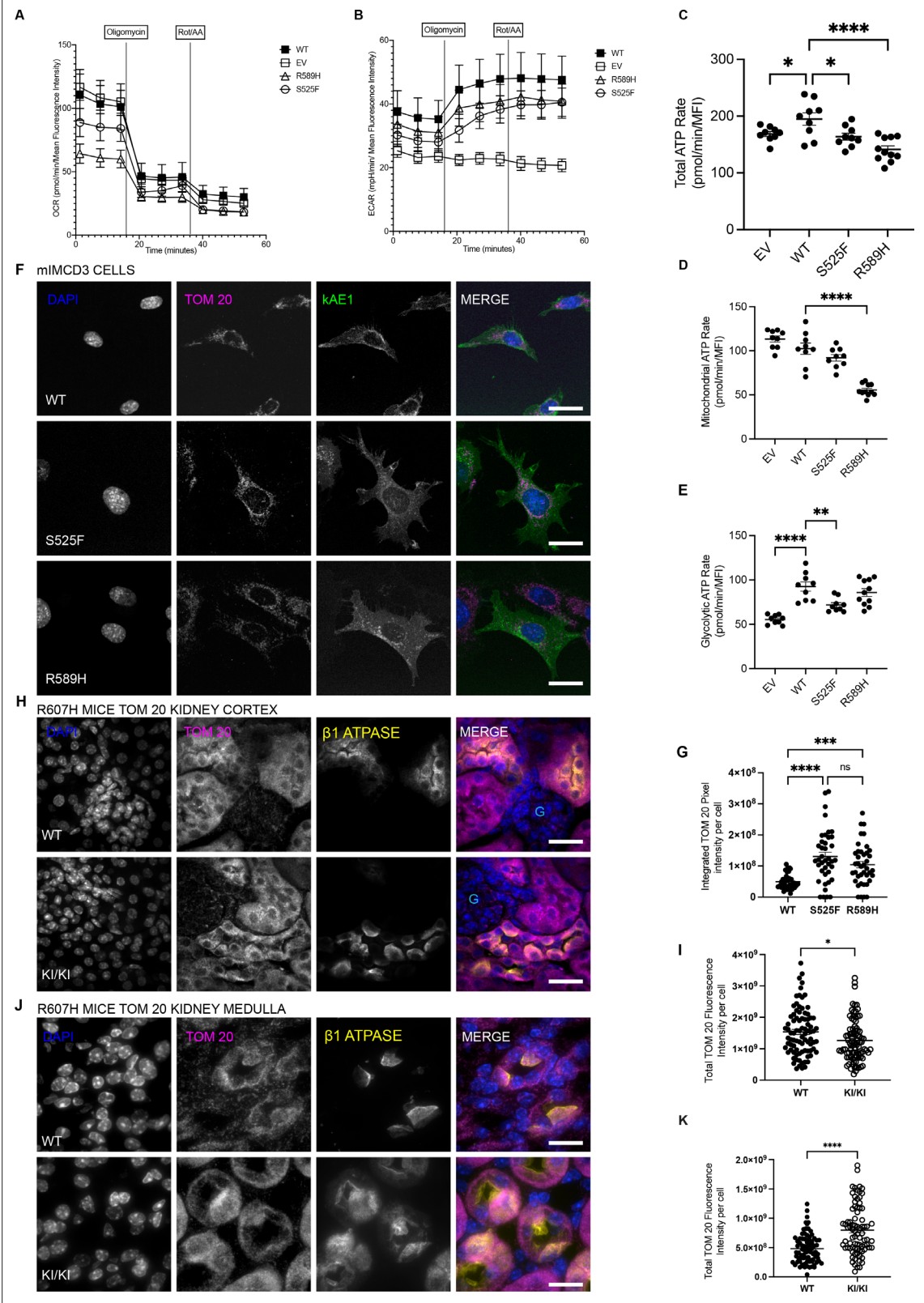

**Figure 5.** dRTA kAE1 mutant cells have lower ATP production rate and abnormal mitochondrial content. Oxygen consumption rate (OCR) (**A**) and extra cellular acidification rate (ECAR) (**B**) of empty vector (EV) transfected cells, kAE1 WT, S525F, or R589H mIMCD3 cells analyzed in a Seahorse XFe96 Extracellular Flux Analyzer with the ATP Rate Assay Test Kit. All cell lines, including EV-transfected cells, were incubated with doxycycline to eliminate a potential effect of doxycycline on measurements. (**C**) Graphical representation of the combination of ATP production rates from mitochondrial

*Figure 5 continued on next page*

*Figure 5 continued*

respiration (mitoATP) and glycolysis (glycoATP) of kAE1 WT, S525F, and R589H mIMCD3 cells measured in real-time following sequential injections of oligomycin and Rotenone + Antimycin-A. Error bars correspond to mean ± SEM, n=minimum. *p<0.05, ****p<0.0001 using one-way ANOVA followed by Tukey's post hoc test. Graphical representations of mitochondrial respiration (**D**) and glycolytic ATP production (**E**) in kAE1 WT, S525F, and R589H mIMCD3 cells. Error bars correspond to mean ± SEM, n=minimum 8. **p<0.01, ****p<0.0001 using one-way ANOVA followed by Tukey's post hoc test. (**F**) Immunofluorescence staining of TOM20 and kAE1 in kAE1 WT, S525F, and R589H mIMCD3 cells. Blue = nuclei, magenta = TOM20, green = kAE1. Scale bar = 8 μm. (**G**) Graphical representation of total TOM20 fluorescence intensity per cell expressing kAE1. Error bars correspond to mean ± SEM, n=minimum 40. ***p<0.001, ****p<0.0001 using one-way ANOVA followed by Tukey's post hoc test. Immunofluorescence images of TOM20 and ß1 ATPase in kidney cortex (**H**) or medulla (**J**) of kAE1 R607H WT and KI mice exposed to a salt-depleted diet with an acid challenge (***Mungara et al., 2024***). Blue = nuclei, magenta = TOM20 (mitochondria), yellow = ß1 ATPase, light blue 'G' indicates the location of a glomerulus. Scale bar = 8 μm. Graphical representation of the total TOM20 fluorescence intensity in ß1 ATPase-positive cells in the cortex (**I**) or medulla (**K**) of the kidney. Error bars correspond to mean ± SEM, n=90. *p<0.05, ****p<0.0001 using Student's *t*-test.

of dRTA in our in vitro and in vivo models involves an inhibition of autophagy flux at the downstream steps involving autophagosome-lysosome fusion and lysosomal protein clearance (***Ballabio and Gieselmann, 2009***; ***Eskelinen, 2006***; ***Mizushima, 2018***; ***de Araujo et al., 2020***).

The question remained as to how these dRTA variants altered autophagy. The *SLC4A1-3* gene family that includes AE1 are regulators of intracellular pH in different cell types (***Thornell and Bevensee, 2015***; ***Zhang et al., 2023***; ***Romero et al., 2013***). The reduced anion exchange activity in mIMCD3 cells expressing the R589H and S525F variants expectedly correlated with an increased pHi compared to WT counterparts. Given that pHi variations can impact autophagy (***Berezhnov et al., 2016***; ***Korolchuk et al., 2011***; ***Heuser, 1989***; ***Xu et al., 2011***; ***Ratto et al., 2022***), we wondered whether this was the case for cells expressing kAE1 dRTA mutants. We observed that an acidic pHi in mutant cells restored autophagy levels similar to WT. Previous studies reported more perinuclear localization of lysosomes and autophagosome-lysosome fusion in cells with an increased pHi (***Korolchuk et al., 2011***; ***Heuser, 1989***). Although not examined in our study, an increase in intracellular pH due to starvation decreased the levels of lysosomal kinesin superfamily member KIF2 and ADP-ribosylation factor-like 8B (ARL8), which are responsible for redistributing lysosomes to the cell periphery. This reduction subsequently inhibited mTORC1, resulting in increased autophagosome synthesis and autophagosome-lysosome fusion (***Korolchuk et al., 2011***).

While an alkaline cytosolic pH partially explains the impairment in autophagy flux, a close relationship also exists between intracellular pH and cellular energy dynamics and metabolic stress, all of which are key regulators of autophagy (***Mizushima and Komatsu, 2011***). Therefore, it was plausible that the kAE1 variant-induced autophagy dysregulation occurs through a signaling pathway akin to that of energy deprivation-induced autophagy. This hypothesis is supported by our findings of reduced ATP production rate in dRTA kAE1 S525F and R589H mutant cells compared to kAE1 WT cells. We also found that both kAE1 S525F and R589H mutant cells have higher mitochondrial content compared to kAE1 WT cells. The increased mitochondrial content coupled with low ATP points towards improper mitochondrial function in dRTA variant cells. Low ATP levels as seen in dRTA mutant cells may impair autophagy, as was shown in human RPE cells (***Schütt et al., 2012***). ATP reduction in RPE cells led to complex mitochondrial changes such as structural disorganization, enzyme activity decline, and oxidative damage to mitochondrial components and DNA (***Schütt et al., 2012***). Similarly, in pancreatic islet cells, an alkaline pHi led to increased uptake of phosphate by mitochondria, accelerating the production of superoxide, promoting mitochondrial permeability transition, and inducing translational attenuation due to endoplasmic reticulum stress, ultimately impairing insulin secretion (***Nguyen et al.,***

**Table 1.** Summary of findings in dRTA kAE1 variant expressing cells compared to WT.

| dRTA variant | Transport activity | Intracellular pH | LC3B accumulation | Autophagy flux | Cellular lysosome size | Cellular lysosome number | Cellular mitochondrial abundance | Rescued autophagy |
|---|---|---|---|---|---|---|---|---|
| *R295H* | Unchanged | Unchanged | Unchanged | N/A | N/A | N/A | N/A | N/A |
| *S525F* | Reduced | Increased | Increased | Blocked downstream | Unchanged | Increased | Increased | Yes |
| *R589H* | Slightly Reduced | Increased | Increased | Blocked downstream | Increased | Unchanged | Increased | Yes |

N/A, not applicable.

**Table 2.** Summary of findings in intercalated cells from dRTA R607H knock-in (KI) relative to WT mice.

| Genotype | LC3B accumulation | Intercalated cell lysosome size | Intercalated cell lysosome number | Intercalated cell mitochondrial abundance |
|---|---|---|---|---|
| *R607H KI/KI* | Increased total LC3B | Increased | Increased | Increased |

*2016*). Overall, our data support that expression of the kAE1 variants increases pHi, which alters mitochondrial function and leads to reduced cellular energy levels that eventually attenuate energy-dependent autophagic pathways including autophagosome-lysosome fusion and lysosomal protein clearance.

In light of these observations, we postulated that correcting the alkaline pHi of dRTA mutant-expressing mIMCD3 cells will alleviate this blockage in autophagy flux. We observed that a chemically engineered pHi of 6.9 (*Lyons et al., 1992*) reduced LC3B II accumulation and LAMP1 abundance in mIMCD3 mutant cells to expression levels similar to that of WT cells at baseline. This suggests that a chemically reduced pHi facilitated protein clearance in the two dRTA mutant cells (*Berezhnov et al., 2016*; *Korolchuk et al., 2011*) as noted in other studies. In one such study, treatment of SH-SY5Y cells with FCCP and nigericin also acidified intracellular pH and triggered autophagy and mitophagy (*Berezhnov et al., 2016*). Similarly, acid loading in proximal tubular cells under chronic metabolic acidosis showed increased autophagic flux and mitophagy (*Namba et al., 2014*). These findings are in line with our results and establish a link between altered autophagy flux and the alkaline pHi of dRTA variant cells. Overall, our study provides one pathway (altered pHi) by which dRTA may arise. However, different variants induce different degrees of functional defects as seen in *Figure 1F & G*. The kAE1 R295H, the only reported amino acid substitution in the amino-terminal cytosol causing dRTA, does not affect the transporter's function or pHi. Therefore, this variant may cause dRTA via a different pathway, for example, defective protein–protein interactions, than transport-defective S525F or partially inactive R589H variants.

In conclusion, our study established a strong relationship between the expression of defective kAE1 proteins, reduced mitochondrial activity, decreased autophagy, and impaired protein degradative flux. Whether this abnormal degradative pathway explains the premature loss of A-IC will need to be elucidated in further studies.

## Materials and methods
### Antibodies and chemicals
Mouse anti-HA antibody (hemagglutinin, Biolegend, formerly Covance), mouse anti-β-actin antibody (Sigma Aldrich or anti-β-actin HRP clone 2F1-1 Biolegend cat#643807), mouse anti-LC3B antibody (Cell Signaling), mouse anti-IVF 12s antibody (Developmental Studies Hybridoma Bank), rat anti-ATP6V1B1 antibody (BiCell #20901), rabbit anti-mTOR antibody and phospho mTOR antibody (Cell Signaling), rabbit anti-4E-BP1 antibody and phospho 4E-BP1 (Ser65) antibody (Cell Signaling), mouse anti-p62 antibody (Abcam), mouse anti-p53 antibody (Cell Signaling), rabbit anti-Cleaved caspase 3 antibody (Cell Signaling), mouse monoclonal anti $Na^+/K^+$-ATPase H-3 antibody (Santa Cruz Biotechnology, Dallas, TX), goat anti-mouse antibody horseradish peroxidase conjugated (HRP) (Cell Signaling Technology), Cy3-conjugated donkey anti-mouse antibody, anti-rabbit and anti-goat antibodies (Jackson Immunoresearch), X-tremeGENE.

### Newly identified *SLC4A1* variations from dRTA patients
The patient carrying the R295H mutation was a boy carrying the variation in the homozygous state, whose genetic diagnosis was made at the age of 5, following growth retardation of –2 SD for both weight and height, with bicarbonate at 18 mmol/L, potassium at 2.8 mmol/L, chloridemia at 94 mmol/L, calcemia at 2.55 mmol/L, and a urinary pH of 7.5. He also had a history of a pyeloureteral junction syndrome that was surgically managed at the age of 3. The R295H dRTA variation is a nonsense homozygous substitution characterized by a replacement of guanine (G) on position 884 by adenine (A) in the coding sequence. It has an allelic frequency of 0.14% in the European population.

For the Y413H variant, the patient was a female diagnosed at 1 month old, with a urinary pH of 7.5, evidence of nephrocalcinosis, and failure to thrive. The S525F variant has been previously reported (*Bertocchio et al., 2020*), but in brief, the patient was a 13-year-old female with plasma pH of 7.25, plasma bicarbonate at 15.3 mmol/L who also presented with polyurethral junction syndrome, nephrocalcinosis, and nephrolithiasis since childhood. The Y413H and S525F dRTA variations are nonsense heterozygous substitutions characterized by a replacement of thymine (T) at position 1237 by cytosine (C) and a replacement of cytosine (C) at position 1574 by thymine (T), respectively. The R589H dRTA variation has been previously described (*Mumtaz et al., 2017*). No follow-up data were available for all patients.

## Mice

Transgenic mice carrying the R607H knockin (KI) mutation (murine equivalent to human R589H mutation) were previously described (*Mumtaz et al., 2017*). Homozygous mice used throughout the study display incomplete dRTA with alkaline urine without metabolic acidosis at baseline as previously reported (*Mumtaz et al., 2017*). Homozygous mice or wild-type (WT) littermates were fed a standard rodent chow (Picolab Rodent Diet 20 # 5053, LabDiet, ST. Louis, MO, USA) or for *Figure 5H-K*, a salt-depleted diet with acid challenge as previously reported (*Mungara et al., 2024*) with adequate and constant water supply, and maintained on a 12hour light and dark cycle throughout their lifespan.

## Cell lines, transfections, and viral infection

Mouse inner medullary collecting duct (mIMCD) cells (ATCC# CRL2123) were used for preparing kAE1 wild type and mutant cell lines. The pLVX TRE3G kAE1 construct was generated from the shuttling of human kAE1 cDNA with an external hemagglutinin (HA) epitope in position 557 (on eAE1) into pLVX-TRE3G plasmid (Clontech) (*Lashhab et al., 2019*). This construct encodes a protein described as kAE1 throughout this paper. The kAE1 S525F and R589H mutants were generated with Q5 site-directed mutagenesis. All plasmids were introduced into the mIMCDs using a viral single-shot packaging kit (Clontech).

## Mouse intercalated cell isolation and tissue homogenate preparation

Kidney tissue homogenates were prepared as previously described (*Mungara et al., 2025*). Briefly, after decapsulation, freshly dissected kidneys were homogenized in cold lysis buffer (0.3 M sucrose, 25 mM imidazole, 1 mM EDTA, 8.5 µM leupeptin, 1 mM PMSF), and vortexed over 1 hour every 15 min. The homogenates were then centrifuged at 14,000 rpm for 15 minutes at 4°C prior to measurement of protein concentration by Bicinchoninic Acid Protein Assay. Primary intercalated cells were prepared from homozygous kAE1 R607H transgenic mice. After cardiac perfusion with PBS, heparin, and collagenase B (Sigma Aldrich), kidneys were homogenized by MACS dissociation and intercalated cells enriched using CD 117 magnetic sorting (Miltenyi Biotec) as previously described (*Saxena et al., 2021*). During the selection, intercalated cells were kept in MACS buffer (PBS, 2 mM EDTA, and 0.5% FBS). Cells were lysed with RIPA lysis buffer (2 mM EDTA, 2% deoxycholate, 0.3 M NaCl, 20 mM Tris/HCl pH 7.5, 2% Triton X-100, 0.2% SDS, pH 7.4), supplemented with complete EDTA-free protease inhibitors, and PhoSTOP phosphatase inhibitor (Roche), PMSF, pepstatin, leupeptin, and aprotinin. An aliquot was saved for bicinchoninic acid assay to determine protein concentration, and remaining lysates were kept in Laemmli buffer at –20°C for immunoblotting.

## Bicarbonate transport assay

This assay has been previously described (*Sterling and Casey, 1999*). Briefly, confluent kAE1 WT-HA or mutant mIMCDs cells grown on coverslips were incubated with 1 µg/mL doxycycline (Sigma-Aldrich) for 18–24 hours at 37°C to induce kAE1 expression. They were then incubated with 2',7'-Bis-(2-Carboxyethyl)–5-(and-6)-Carboxyfluorescein, Acetoxymethyl Ester (BCECF–AM, Thermo Scientific), a fluorophore which excites at 440 and 490 nm and emits 510 nm wavelength for 30 minutes at 37°C. Using a fluorometer from Photon Technologies International (PTI) (London, Ontario, Canada), coverslips were perfused with NaCl-based Ringer's buffer (5 mM glucose, 5 mM potassium gluconate, 1 mM calcium gluconate, 1 mM magnesium sulfate, 10 mM HEPES, 2.5 mM sodium dihydrogen phosphate, 25 mM sodium bicarbonate, 140 mM sodium chloride) for 5–10 minutes. Once stable, initial fluorescence (corresponding to steady-state pHi) was recorded for the first 30 seconds before switching to

chloride-free containing sodium gluconate-based Ringer's buffer of same osmolality. BCECF fluorescence was calibrated by perfusing cells with different pH buffers (6.5, 7, 7.5) in the presence of 10 mM nigericin. The Ringer's buffers were continuously bubbled with an air:$CO_2$ mixture (19:1), providing 5% $CO_2$. Transport rates of the cells were determined by linear regression of the initial fluorescence variations (over the first 60 seconds), normalized to pH calibration measurements. All measurements were done using PTI FelixGX software.

## Cell treatments and immunoblotting

For autophagy experiments, kAE1 WT or mutant cells were seeded to 70% confluency on 10 cm culture plates and treated with 1 µg/mL doxycycline (Sigma-Aldrich) for 48 hours. Cells were then either treated with 400 nM bafilomycin A1 for 4 hours to inhibit autophagy or with Hanks balanced salt solution (HBSS, Gibco) to starve cells and induce autophagy or given no treatment. To chemically modify pHi of cells, 90–100% confluent cells were treated with 1 µg/mL of doxycycline (Sigma-Aldrich) overnight. Cells were then incubated in Ringer's buffer with pH 6.6, supplemented with 0.03 µM nigericin with a final potassium concentration of 168 mM for 2 hours at 37°C. Steady-state cells were incubated in normal pH media without nigericin prior to lysis. Under treated conditions, pHi was similar to pHe (*Figure 3—figure supplement 1*). Cells were lysed with RIPA lysis buffer (1% deoxycholate, 1 mM EDTA, 0.15 M NaCl, 0.1% SDS, 10 mM TRIS/HCl [pH 7.5], 1% Triton X-100) with phosphatase inhibitors (cat. no. 04906837001; Roche PhosSTOP) and protease inhibitors (cat. no. 04693159001; Roche Complete Tablets, Mini EDTA-free) and stored at −20°C with or without 2x Laemmli buffer. The aliquot without Laemmli buffer was used for a bicinchoninic acid (BCA) assay to determine protein concentration. Following the BCA, 10–30 mg of total protein was loaded on SDS-PAGE gels. Proteins were transferred to PVDF membranes and antibodies listed above were used for detecting the proteins of interest. Primary antibodies were diluted in 1% milk and incubated on membranes overnight at 4°C followed by secondary antibodies linked with horseradish peroxidase (HRP) for 1 hour at room temperature. Protein detection was done with the Enhanced Chemiluminescence reagent (ECL Prime, Invitrogen), and a BioRad Imager. The ImageLab software (BioRad) was utilized for the quantification of relative band intensities.

## Cell surface biotinylation assay

mIMCD3 cells stably expressing kAE1 WT, S525F, or R589H were seeded to 70–80% confluency. The cells were incubated with sulfo-N-hydroxysuccinimide-SS-biotin (Thermo cat.# 21331) (1.5 mg/mL in ice-cold PBS) for an hour at 4°C, quenched with 100 mM glycine and lysed with RIPA lysis buffer, supplemented with complete EDTA-free protease inhibitors, and PhoSTOP phosphatase inhibitor (Roche), PMSF, pepstatin, leupeptin, and aprotinin. Total protein levels were measured by BCA, and an aliquot was saved as 'Total' fraction. 450 mg of each lysate was subsequently incubated with 100 µL streptavidin slurry beads for 1 hour on a rocker at 4°C. Following centrifugation, the supernatant was collected, and an aliquot kept as the 'unbiotinylated' fraction. After six washes, the beads were resuspended in 50 µL of 2X Laemmli buffer and incubated at room temperature for 30 minutes. The eluted biotinylated proteins were subsequently collected by centrifugation ('Biotinylated' fraction). The biotinylated fraction (45 µL) was loaded on SDS-PAGE gel for immunoblot analysis along with 3 µg of the Total fraction and a matched volume of unbound fraction per well. In addition to anti-HA antibody, the blots were probed for actin to ensure cell membrane integrity was intact during the biotinylation procedure, and for Na$^+$/K$^+$-ATPase as cell surface control.

## Magic Red assay

80% confluent kAE1 WT or mutant mIMCD3 cells were treated with 1 µg/mL doxycycline (Sigma-Aldrich) for 48 hours to induce kAE1 expression. Different treatments, including a 4-hour incubation with 400 nM bafA1 to inhibit autophagy, a 2-hour starvation in HBSS to induce autophagy, or no treatment (steady state), were applied. Cells were then incubated with 1% Magic Red reagent (ImmunoChemistry Technologies) in DMEM-F12 medium at 37°C in the dark for 30–60 minutes. Cells were then fixed with 4% PFA, quenched with 100 mM glycine, permeabilized with 0.2% Triton X-100, blocked with 1% BSA, and incubated with mouse anti-HA primary antibody and donkey anti-mouse Alexa Fluor 488 conjugated secondary antibody for 30 minutes each. Cells were then incubated with 4′,6-diamidino-2-phenylindole (DAPI) for 5 minutes before mounting using DAKO Mounting Medium

(Agilent Technologies). A WaveFX confocal microscope was used to image the slides, and the images were analyzed blindly using the Fiji software.

## Autophagy flux assay

kAE1 WT-HA or mutant mIMCD3 cells seeded to 70% confluency in a 6-well plate on coverslips were transiently transfected with the eGFP-RFP-LC3 cDNA construct (kind gift from Dr. Goping, Department of Biochemistry, University of Alberta) using the X-tremeGENE HP DNA transfection reagent (Roche). 4 hours after transfection, the cells were incubated with 1 µg/mL doxycycline for 48 hours at 37°C to induce kAE1 expression. Following this incubation, cells were incubated with blocking medium, mouse anti-HA primary antibody (1:200), and donkey anti-mouse Alexa Fluor 649 (Jackson ImmunoResearch). Hoechst stain (ImmunoChemistry Technologies) was used to stain cellular nuclei. A WaveFX confocal microscope together with the Velocity and Fiji software was used to capture and analyze images.

## Assessment of mitochondrial content

kAE1 WT-HA or mutant mIMCD3 cells seeded to 50% confluency in a 6-well plate on coverslips were incubated with 1 µg/mL doxycycline for 48 hours at 37°C and overnight in complete medium with no antibiotics. Cells were then fixed with 4% PFA, quenched with 100 mM glycine, permeabilized with 0.2% Triton X-100, blocked with 1% BSA, and incubated with mouse anti-HA and rabbit anti-TOM20 primary antibodies and then with donkey anti-mouse Alexa Fluor 488 and anti-rabbit Cy3 conjugated secondary antibodies. Cells were then incubated with DAPI for 5 minutes before mounting using DAKO Mounting Medium (Agilent Technologies). A WaveFX confocal microscope was used to image the slides, and the images were analyzed blindly using the Fiji software.

## Metabolic flux analysis

kAE1 WT-HA or mutant mIMCD3 cells were treated with 1 µg/mL doxycycline for 48 hours followed by an overnight incubation in complete media with no antibiotics and cells seeded at $2 \times 10^4$ cells per well in Seahorse XFe 96-well plates overnight to form a uniform monolayer. On the day of assay, culture medium was replaced with XF DMEM Medium pH 7.4 (103575-100, Agilent Technologies) with glucose (10 mM), pyruvate (1 mM), and L-glutamine (2 mM) and incubated in a non-$CO_2$, 37°C incubator for 1 hour prior to their placement into the XFe96 Analyzer. Using the ATP production rate assay kit (#103592-100, Agilent Technologies) and XF cell Mito Stress Test kit (#103015-100, Agilent Technologies), metabolic indices were obtained from the Seahorse XFe96 Analyzer following manufacturer's procedures previously described (*Sawasdee et al., 2010*). The total ATP rate is the sum of ATP production rate from both glycolysis and oxidative phosphorylation. Glycolysis releases protons in a 1:1 ratio with ATP; hence, the glycolytic ATP rate is calculated from the glycolytic proton efflux rate (glycoPER). GlycoPER is determined by subtracting respiration-linked proton efflux from total proton efflux by inhibiting complex I and III. The empty vector transfected cells provided a control for a potential effect of doxycycline on measurements. Oxygen consumption rate (OCR) and extracellular acidification rate (ECAR) were measured at various time points at basal state followed by injections of oligomycin (1.5 µM) and Rotenone + Antimycin A (0.5 µM).

## Tissue preparation and immunostaining of kidney sections

Kidneys collected after perfusion were stored in 4% PFA overnight at 4°C. The PFA solution was switched to 15% sucrose for 2 hours and then transferred to 30% sucrose solution overnight at 4°C. Thereafter, kidneys were fixed in O.C.T (Tissue-Tek) and snap frozen in liquid nitrogen. These tissues were stored at –80°C until cryo-sectioning. Ten (10) micron tissue sections were fixed on a charged glass slide (Thermo Fisher) and used immediately or stored at –80°C until immunostaining. For immunostaining, the slices were first air-dried for 20 minutes, washed with PBS for 5 minutes, and fixed with 4% PFA for 20 minutes at 4°C. The sections were quenched with 100 mM glycine for 15 minutes, permeabilized, and blocked with 5% or 10% serum in 0.2% Triton in PBS for 1 hour at room temperature. Slices were incubated in primary antibody diluted in 5% or 10% serum overnight at 4°C followed by secondary antibody diluted in 5% or 10% serum for 1 hour at room temperature. Slices were washed with 0.1% Tween 20 in PBS and incubated with DAPI for 5 minutes at room temperature, mounted with DAKO mounting medium, and sealed. Slides were air-dried and then stored at –20°C.

Note that kidney sections analyzed in *Figure 5H–K* were obtained from mice fed a 'salt-depletion with acid load' diet consisting of a low sodium and chloride diet for 8 days, complemented with 0.28 M $NH_4Cl$ with 0.5% sucrose in drinking water for six additional days as previously described (*Escobar et al., 2016*). This diet triggered a metabolic acidosis, significantly lower plasma bicarbonate with a more alkaline urine in the homozygous KI mice compared to WT littermates.

### Confocal imaging and image analysis

Immunofluorescent imaging was done with a WaveFX confocal microscope (Quorum Technologies, Guelph, Ontario, Canada) powered by a Volocity software (Quorum Technologies). Images were taken with ×40 oil immersion objective with z-stacks at 0.5 µm intervals. Quantitative image analysis was performed using the Volocity analysis software or by open-source cell image analysis software CellProfiler (*Stirling et al., 2021*) and Fiji (*Schindelin et al., 2012*).

### Image analysis

**Cell profiler** was used to analyze TOM20 staining in mIMCD3 cells. After converting the three-channel RGB images to grayscale using the split method, we manually outlined each cell in the channel corresponding to kAE1 staining and inputted it back into the pipeline. The objects were then converted to a binary image and the TOM20 channel was used as the input channel to measure intensity of TOM20 staining in the manually outlined objects. The raw data, including the sum of TOM20 pixel intensity per cell, were then analyzed using GraphPad Prism software. A minimum of 30 cells were analyzed.

 **Fiji** software was used to analyze images from Magic Red staining and autophagy flux experiment. After each multichannel image was opened and merged in Fiji, the 'multi point' tool was used to label all regions of interest. Freehand selection tool was used to manually draw the outline of all selected regions of interest, followed by the 'Analyze' command to extract number and size of puncta in pixels. Pixel values were then converted to microns and data analysis was completed using GraphPad Prism. A minimum of 30 cells was analyzed.

 **Volocity** software was used for analysis of kidney sections. From B1 $H^+$-ATPase-positive single cells cropped from confocal images of medullary or cortical mouse kidney sections, the channel of interest was used to find objects using the 'Find objects' command. Refinement within the selection was made based on object size. The minimum object size thresholds were 0.016 $µm^2$ for LAMP1 puncta and 0.02 $µm^2$ for TOM20 staining. This was then labeled as 'population one'. From population one, touching objects were separated using a size guide of 0.02 $µm^2$ for LAMP1 puncta only with the 'Separate touching objects' function. For TOM20 staining, all RFP-positive stain within individual cells was characterized under one mask without separating touching objects. The minimum object size was set to 0.02 $mm^2$ and everything less than that was considered background staining. The sum of TOM20 fluorescence intensity per cell was collected and analyzed using GraphPad Prism. For LAMP1 images, an exclusion criterion removing objects lesser than or equal to 0.02 $mm^2$ was used to remove small background objects. Once the regions of interest in the image were properly outlined, data, including number of puncta, intensity of puncta, area/ volume of puncta, among other measurements, were exported and data analyzed with GraphPad Prism. A minimum of 60 cells was analyzed.

### Statistical analysis

All the experiments were independently repeated a minimum of three times. Experimental results were analyzed using the GraphPad Prism software and are summarized as mean ± SEM. All statistical comparisons were made using unpaired Student's *t*-test or one/two-way ANOVA followed by a post hoc test as indicated in figure legends. A p-value $<0.05$ was considered statistically significant. All datasets were assessed for normality, and outliers identified by Prism were excluded.

### Acknowledgements

We thank Kristina MacNaughton, Jared Bouchard, Kiera Smith, and Hilmar Strickfaden for excellent technical assistance. Imaging experiments were performed at the University of Alberta Faculty of Medicine & Dentistry Cell Imaging Core, RRID:SCR_019200, which receives financial support from the Faculty of Medicine & Dentistry, the University Hospital Foundation, Striving for Pandemic Preparedness – The Alberta Research Consortium, and Canada Foundation for Innovation (CFI) awards to contributing investigators. Services were provided by the University of Alberta Faculty of Medicine &

Dentistry Workshop, RRID:SCR_019181, which receives financial support from the Faculty of Medicine & Dentistry. This study was funded by the Canadian Institutes of Health Research (PJT#168871) and the Kidney Foundation of Canada (2020KHRG-666615) to EC, by a grant from the Deutsche Forschungsgemeinschaft to MJS (IRTG 1830), and operating funds from Natural Sciences and Engineering Research Council of Canada (RGPIN-2018–05783) and the Canadian Institutes of Health Research (PS#165816) to NT. GE received a Graduate Student Engagement Scholarship, a Faculty of Medicine and Dentistry Delnor Scholarship, and a Faculty of Medicine and Dentistry 75th Anniversary award. MR received a Sir Frederick Banting and Dr. Charles Best Canada Graduate Scholarship-Master's (CGS-M) from the Canadian Institutes of Health Research; Walter H Johns Graduate Fellowship; a University of Alberta Faculty of Medicine and Dentistry/Alberta Health Services Graduate Student Recruitment Studentship (GSRS) and an Alberta Graduate Excellence Scholarship (AGES). AKMSU received an NSERC CREATE graduate studentship. FC was supported by a Discovery Grant to EC from the Natural Sciences and Engineering Research Council (RGPIN-2017-06432), and was awarded a Graduate Recruitment scholarship from the University of Alberta. SMAH received a PhD scholarship from the DAAD (Deutscher Akademischer Austauschdienst).

## Additional information

### Funding

| Funder | Grant reference number | Author |
| --- | --- | --- |
| Canadian Institutes of Health Research | PJT#168871 | Emmanuelle Cordat |
| Natural Sciences and Engineering Research Council of Canada | RGPIN-2017-06432 | Forough Chelangarimiyandoab |
| Kidney Foundation of Canada | 2020KHRG-666615 | Emmanuelle Cordat |
| Deutsche Forschungsgemeinschaft | IRTG1830 | Manfred J Schmitt |
| Deutscher Akademischer Austauschdienst | PhD Scholarship | Sarder MA Hasib |
| Canadian Institutes of Health Research | PS165816 | Nicolas Touret |
| Natural Sciences and Engineering Research Council of Canada | RGPIN-2018-05783 | Nicolas Touret |
| University of Alberta | Multiple Scholarships | Grace Essuman Midhat Rizvi Forough Chelangarimiyandoab Priyanka Mungara |
| Canadian Institutes of Health Research | Canada Graduate Scholarship-Master's | Midhat Rizvi |
| Natural Sciences and Engineering Research Council of Canada | CREATE Graduate Studentship | Shahid AKM Ullah |

The funders had no role in study design, data collection and interpretation, or the decision to submit the work for publication.

### Author contributions

Grace Essuman, Conceptualization, Data curation, Formal analysis, Validation, Investigation, Writing – original draft, Writing – review and editing; Midhat Rizvi, Sarder MA Hasib, Data curation, Formal analysis, Validation, Investigation, Writing – review and editing; Ensaf Almomani, Data curation, Formal analysis, Validation, Investigation, Methodology, Writing – review and editing; Shahid AKM

Ullah, Priyanka Mungara, Data curation, Formal analysis, Validation, Methodology, Writing – review and editing; Forough Chelangarimiyandoab, Data curation, Formal analysis, Methodology, Writing – review and editing; Manfred J Schmitt, Supervision, Funding acquisition, Validation, Writing – review and editing; Marguerite Hureaux, Rosa Vargas-Poussou, Data curation, Formal analysis, Funding acquisition, Validation, Writing – review and editing; Nicolas Touret, Formal analysis, Supervision, Investigation, Methodology, Writing – review and editing; Emmanuelle Cordat, Conceptualization, Resources, Formal analysis, Supervision, Funding acquisition, Validation, Investigation, Visualization, Methodology, Writing – original draft, Project administration, Writing – review and editing

### Author ORCIDs
Grace Essuman ⬤ https://orcid.org/0000-0003-3002-7880
Forough Chelangarimiyandoab ⬤ https://orcid.org/0000-0002-0255-2242
Nicolas Touret ⬤ https://orcid.org/0000-0003-3700-6302
Emmanuelle Cordat ⬤ https://orcid.org/0000-0001-9875-5804

### Ethics
This study was conducted in accordance with all national and institutional animal care guidelines and approved by the University of Alberta's Animal Care and Use Committee (AUP #1277).

Reviewer #1 (Public review): https://doi.org/10.7554/eLife.108253.3.sa1
Reviewer #2 (Public review): https://doi.org/10.7554/eLife.108253.3.sa2
Reviewer #3 (Public review): https://doi.org/10.7554/eLife.108253.3.sa3
Author response https://doi.org/10.7554/eLife.108253.3.sa4

## Additional files

### Supplementary files
MDAR checklist

### Data availability
Source Data (including Western Blot source data) are available at: https://doi.org/10.5061/dryad.2bvq83c4f.

The following dataset was generated:

| Author(s) | Year | Dataset title | Dataset URL | Database and Identifier |
|---|---|---|---|---|
| Essuman G, Rizvi M, Almomani E, Ullah SAKM, Hasib SMA, Chelangarimiyandoab F, Mungara P, Schmitt MJ, Hureaux M, Vargas-Poussou R, Touret N, Cordat E | 2026 | Data: SLC4A1 mutations that cause distal renal tubular acidosis alter cytoplasmic pH and cellular autophagy | https://doi.org/10.5061/dryad.2bvq83c4f | Dryad Digital Repository, 10.5061/dryad.2bvq83c4f |

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
