## [Editor Report · eLife Assessment]

This work reports the characterization of newly identified genetic variants of SLC4A1 in patients with distal renal tubular acidosis. Cell culture studies supplemented with histological analysis of a previously established disease mouse model provide **convincing** evidence that some of the variants increase intracellular pH, reduce ATP synthesis, and attenuate autophagic degradative flux. The study is **valuable** in establishing a mechanistic framework for future exploration of the link between intracellular pH and mutations in SLC4A1 in vivo.

---

## [Referee Report · Reviewer #1 (Public review)]

Summary:

This study is an evaluation of patient variants in the kidney isoform of AE1 linked to distal renal tubular acidosis. Drawing on observations in the mouse kidney, this study extends findings to autophagy pathways in a kidney epithelial cell line.

Strengths:

Experimental data are convincing and nicely done.

The revised manuscript incorporates most of the reviewer recommendations and presents a more cohesive story that is easier to read and assess. The data are convincing, of suitable quality and nicely presented. Statistical evaluation is rigorous. The link between kAE1 mutants and cell metabolism and autophagy is novel and provides insights on pathological observations in dRTA.

---

## [Referee Report · Reviewer #2 (Public review)]

Context and significance:

Distal renal tubular acidosis (dRTA) can be caused by mutations in a Cl-/HCO3- exchanger (kAE1) encoded by the SLC4A1 gene. The precise mechanisms underlying the pathogenesis of the disease due to these mutations is unclear, but it is thought that loss of the renal intercalated cells (ICs) that express kAE1 and/or aberrant autophagy pathway function in the remaining ICs may contribute to the disease. Understanding how mutations in SLC4A1 affect cell physiology and cells within the kidney, a major goal of this study, is an important first step to unraveling the pathophysiology of this complex heritable kidney disease.

Summary:

The authors identify a number of new mutations in the SLC4A1 gene in patients with diagnosed dRTA that they use for heterologous experiments in vitro. They also use a dRTA mouse model with a different SLC4A1 mutation for experiments in mouse kidneys. Contrary to previous work that speculated dRTA was caused mainly by trafficking defects of kAE1, the authors observe that their new mutants (with the exception of Y413H) traffic and localize at least partly to the basolateral membrane of polarized heterologous mIMCD3 cells, an immortalized murine collecting duct cell line. They go on to show that the remaining mutants induce abnormalities in the expression of autophagy markers and increased numbers of autophagosomes, along with an alkalinized intracellular pH. They also reported that cells expressing the mutated kAE1 had increased mitochondrial content coupled with lower rates of ATP synthesis. The authors also observed a partial rescue of the effects of kAE1 variants through artificially acidifying the intracellular pH. Taken together, this suggests a mechanism for dRTA independent of impaired kAE1 trafficking and dependent on intracellular pH changes that future studies should explore.

Strengths:

The authors corroborate their findings in cell culture with a well characterized dRTA KI mouse and provide convincing quantification of their images from the in vitro and mouse experiments. The data largely support the claims as stated. Some of the mutants induce different strengths of effects on autophagy and the various assays than others, and it is not clear why this is from the data in the manuscript. The authors provide discussion of potential reasons for these differences that future studies could explore.

Weaknesses:

The pH effects of their mutants are only explored in vitro, and the in vitro system has a number of differences from a living mouse kidney or ex vivo kidney slice.

---

## [Referee Report · Reviewer #3 (Public review)]

Summary:

The authors have identified novel dRTA causing SLC4A1 mutations and studied the resulting kAE1 proteins to determine how they cause dRTA. Based on a previous study on mice expressing the dRTA kAE1 R607H variant, the authors hypothesize that kAE1 variants cause an increase in intracellular pH which disrupts autophagic and degradative flux pathways. The authors clone these new kAE1 variants and study their transport function and subcellular localization in mIMCD cells. The authors show increased abundance of LC3B II in mIMCD cells expressing some of the kAE1 variants, as well as reduced autophagic flux using eGFP-RFP-LC3. These data, as well as the abundance of autophagosomes, serve as the key evidence that these kAE1 mutants disrupt autophagy. Furthermore, the authors demonstrate that decreasing the intracellular pH abrogates the expression of LC3B II in mIMCD cells expressing mutant SLC4A1. Lastly, the authors argue that mitochondrial function, and specifically ATP synthesis, is suppressed in mIMCD cells expressing dRTA variants and that mitochondria are less abundant in AICs from the kidney of R607H kAE1 mice. Overall, the authors provide evidence about how new kAE1 mutants may cause dRTA.

Strengths:

The authors cloned novel dRTA causing kAE1 mutants into expression vectors to study the subcellular localization and transport properties of the variants. The immunofluorescence images are generally of high quality and the authors do well to include multiple samples for all of their western blots.

---

## [Author Response]

The following is the authors’ response to the original reviews.

**Public Reviews:**

**Reviewer #1 (Public review):**
Summary:This study is an evaluation of patient variants in the kidney isoform of AE1 linked to distal renal tubular acidosis. Drawing on observations in the mouse kidney, this study extends findings to autophagy pathways in a kidney epithelial cell line.Strengths:Experimental data are convincing and nicely done.

Thank you

Weaknesses:Some data are lacking or not explained clearly. Mutations are not consistently evaluated throughout the study, which makes it difficult to draw meaningful conclusions.

We have revised our manuscript to clarify some earlier explanations and provided rationale for focusing on specific variants throughout the study.

**Reviewer #2 (Public review):**
Context and significance:Distal renal tubular acidosis (dRTA) can be caused by mutations in a Cl-/HCO3- exchanger (kAE1) encoded by the SLC4A1 gene. The precise mechanisms underlying the pathogenesis of the disease due to these mutations are unclear, but it is thought that loss of the renal intercalated cells (ICs) that express kAE1 and/or aberrant autophagy pathway function in the remaining ICs may contribute to the disease. Understanding how mutations in SLC4A1 affect cell physiology and cells within the kidney, a major goal of this study, is an important first step to unraveling the pathophysiology of this complex heritable kidney disease.Summary:The authors identify a number of new mutations in the SLC4A1 gene in patients with diagnosed dRTA that they use for heterologous experiments in vitro. They also use a dRTA mouse model with a different SLC4A1 mutation for experiments in mouse kidneys. Contrary to previous work that speculated dRTA was caused mainly by trafficking defects of kAE1, the authors observe that their new mutants (with the exception of Y413H, which they only use in Figure 1) traffic and localize at least partly to the basolateral membrane of polarized heterologous mIMCD3 cells, an immortalized murine collecting duct cell line. They go on to show that the remaining mutants induce abnormalities in the expression of autophagy markers and increased numbers of autophagosomes, along with an alkalinized intracellular pH. They also reported that cells expressing the mutated kAE1 had increased mitochondrial content coupled with lower rates of ATP synthesis. The authors also observed a partial rescue of the effects of kAE1 variants through artificially acidifying the intracellular pH. Taken together, this suggests a mechanism for dRTA independent of impaired kAE1 trafficking and dependent on intracellular pH changes that future studies should explore.Strengths:The authors corroborate their findings in cell culture with a well-characterized dRTA KI mouse and provide convincing quantification of their images from the in vitro and mouse experiments

Thank you

Weaknesses:The data largely support the claims as stated, with some minor suggestions for improving the clarity of the work. Some of the mutants induce different strengths of effects on autophagy and the various assays than others, and it is not clear why this is from the present manuscript, given that they propose pHi and the unifying mechanism

We have modified our manuscript to discuss the various strengths of the mutants and emphasize that alteration of cytosolic pH by kAE1 variants may not be the only mechanism leading to dRTA.

**Reviewer #3 (Public review):**
Summary:The authors have identified novel dRTA causing SLC4A1 mutations and studied the resulting kAE1 proteins to determine how they cause dRTA. Based on a previous study on mice expressing the dRTA kAE1 R607H variant, the authors hypothesize that kAE1 variants cause an increase in intracellular pH, which disrupts autophagic and degradative flux pathways. The authors clone these new kAE1 variants and study their transport function and subcellular localization in mIMCD cells. The authors show increased abundance of LC3B II in mIMCD cells expressing some of the kAE1 variants, as well as reduced autophagic flux using eGFP-RFP-LC3. These data, as well as the abundance of autophagosomes, serve as the key evidence that these kAE1 mutants disrupt autophagy. Furthermore, the authors demonstrate that decreasing the intracellular pH abrogates the expression of LC3B II in mIMCD cells expressing mutant SLC4A1. Lastly, the authors argue that mitochondrial function, and specifically ATP synthesis, is suppressed in mIMCD cells expressing dRTA variants and that mitochondria are less abundant in AICs from the kidney of R607H kAE1 mice. While the manuscript does reveal some interesting new results about novel dRTA causing kAE1 mutations, the quality of the data to support the hypothesis that these mutations cause a reduction in autophagic flux can be improved. In particular, the precise method of how the western blots and the immunofluorescence data were quantified, with included controls, would enhance the quality of the data and offer more supportive evidence of the authors' conclusions.Strengths:The authors cloned novel dRTA causing kAE1 mutants into expression vectors to study the subcellular localization and transport properties of the variants. The immunofluorescence images are generally of high quality, and the authors do well to include multiple samples for all of their western blots.

Thank you

Weaknesses:Inconsistent results are reported for some of the variants. For example, R295H causes intracellular alkalinization but also has no effect on intracellular pH when measured by BCECF. The authors also appear to have performed these in vitro studies on mIMCD cells that were not polarized, and therefore, the localization of kAE1 to the basolateral membrane seems unlikely, based upon images included in the manuscript. Additionally, there is no in vivo work to demonstrate that these kAE1 variants alter intracellular pH, including the R607H mouse, which is available to the authors. The western blots are of varying quality, and it is often unclear which of the bands are being quantified. For example, LAMP1 is reported at 100kDa, the authors show three bands, and it is unclear which one(s) are used to quantify protein abundance. Strikingly, the authors report a nonsensical value for their quantification of LCRB II in Figure 2, where the ratio of LCRB II to total LCRB (I + II) is greater than one. The control experiments with starvation and bafilomyocin are not supportive and significantly reduce enthusiasm for the authors' findings regarding autophagy. There are labeling errors between the manuscript and the figures, which suggest a lack of vigilance in the drafting process.

The R295H variant was identified in a dRTA patient and as such, it was important to report it. However, this is the first mutation located in the amino-terminus of the protein, which may be involved in protein-protein interactions, so other mechanisms may cause dRTA for this variant. We have therefore modified our manuscript to state that alteration of cytosolic pH may not be the only mechanism leading to dRTA. At this time, we are not able to measure cytosolic pH in vivo and hope to be able to do it in the future.

In our revised manuscript, we also show cell surface biotinylation results supporting that plasma membrane abundance of the kAE1 S525F and R589H variants is not significantly different than WT in non-polarized mIMCD3 cells (Figure 3 A&B), in line with the predominant basolateral localization of the variants in polarized cells (Figure 1C). Therefore, these two mutant proteins are not mis-trafficked in non-polarized cells. Finally, we have clarified which bands have been used for quantification and corrected quantifications (including ratio measurements).

**Recommendations for the authors:**

**Reviewer #1 (Recommendations for the authors):**
(1) R295H is recessively inherited, whereas Y413H is dominantly inherited: this is interesting and may be linked to their cellular expression and function. Is this information known for the other mutations examined in this study?

The S25F and R589H dRTA variants have both been reported to exhibit autosomal dominant inheritance. This information is now updated in lines 146 and 158-159.

(2) R589H expression levels are evaluated in the Western blot of Figure 1, but localization and activity are not examined in Figure 2. However, R589H is included in autophagy experiments shown in later figures. Similarly, mutant R607H is the subject of several experiments further into the manuscript, but no initial analysis is provided for this variant.

Protein abundance and localization of the R589H mutant in mIMCD3 cells have been shown in our previous publication in Supplementary Fig 5D and Supplementary Fig 2J [1]. This now indicated on lines 158-159. Our previous paper also presented a detailed study of the R607H dRTA mutant, the mouse model corresponding to the human R589H mutation. This is now indicated on lines 70, 118-119 and 180. The present study builds upon those published findings.

(3) This inconsistency is confusing, detracts from the usefulness of the study, and makes the comparative analysis of mutations incomplete. It is difficult to extrapolate from published studies in MDCK1 cells, which show different results on trafficking.

The mIMCD3 cell line, which more closely resembles the physiology of the mouse collecting duct than MDCK cells, was selected for this study and our previous one [1]. Accordingly, the results obtained are better aligned with in vivo evidence. In contrast, differences in mutant protein expression and localization observed in other cell lines, like the MDCK cells, are likely attributable to differences in their cellular origin.

(4) In Figure 2, could the authors explain why total LC3B is graphed for the data shown in mouse lysates, whereas the ratio of bands is analysed for cell lysates? Both sets of data show the two LC3B bands.

Total LC3B levels were significantly increased in the mutant compared to WT; however, no significant difference was observed in the lipidation ratio. For this reason, that graph is not shown in the main paper but has been included in the Supplementary Figure 1D.

(5) In Figure 3, representative fluorescence images should be shown for all cell lines.

We have now included representative immunofluorescence images for all cell lines in Figure 3C.

(6) pH effects: Suggest that steady state pHi (Figure 3E) and rate of alkalization (Figure 1F) would be more effective together in Figure 1. The authors should show data for the effect of nigericin on cytoplasmic pH in Figure 3. If the rate of alkalinization in the mutant cells is reduced, shouldn't the intracellular steady state pH be more acidic? A cartoon depicting the transporter activity in the cell and the expected changes in pHi would be helpful. Is there a way to activate/inhibit NHE1 and rescue the effect of the mutant kAE1? It is unclear if the link between the mutant kAE1 and mitochondrial ATP production is a consequence of the intracellular pH or an indirect effect.

We opted to keep the effect of nigericin on pHi in Supplementary Fig1A given that Figure 3 already contains 11 panels. Also, in intercalated cells, the kAE1 protein physiologically exports 1 molecule of bicarbonate in exchange of 1 chloride ion import hence a reduced transport activity would result in a more alkaline intracellular pH. To clarify this point, we have included a diagram in Figure 1E as suggested. However, to calculate the rate of intracellular alkalinisation, the transporter is functioning in the opposite direction, i.e. extruding chloride and importing bicarbonate (see methods protocol for transport assay). Therefore, in this assay (Figure 1G), a defective chloride/bicarbonate activity results in a reduced rate of intracellular alkalinisation rate. This is now explained on lines 169-172.

Disruption of NHE1 function would impair sodium homeostasis and as such, potentially affect the activity of other proteins associated with acid-base balance and autophagy in collecting duct cells. Therefore, any resulting effects may not be confidently attributed specifically to the mutant kAE1. With nigericin, we aimed to alter pHi while affecting the least possible other ion concentration. Due to space considerations, Figure 1 has been reorganised to include the rate of alkalinisation and pHi (panels F and G).

**Reviewer #2 (Recommendations for the authors):**
(1) The authors could improve the readability of this manuscript for a general audience by clarifying and summarizing the respective phenotype(s)/effect(s) of the different mutants in some kind of table in the main figures. It is hard to keep track of the different disease mutants alongside the KI mouse mutations, as the text frequently discusses multiple mutants at a time.

As requested, we added two tables (Supplementary Tables 1 & 2) in Supplementary files summarizing the data obtained in this study. We hope this will help the readership to keep track of each variant’s phenotype.

(2) The subtitle of the results section of Figure 2 should be reworded to reflect that whole kidney lysates are used for the KI mice and not the other mutants.

As requested, the title in the Results section has been modified (lines 178-179).

(3) More discussion of why the different mutants cause different strengths of phenotypes should be included.

Different variants induce different degree of functional defects as seen in Figure 1F & G. The kAE1 R295H, the only amino acid substitution in the amino-terminal cytosol causing dRTA, does not affect the transporter’s function or cells’ pHi. Therefore, this variant may cause dRTA via a different pathway than transport-defective S525F or partially inactive R589H variants that both affect pHi. Our study does not exclude that dRTA may be caused by other defects than pHi alterations, including defective proteinprotein interactions. This discussion is now included in the manuscript on lines 386-391.

**Reviewer #3 (Recommendations for the authors):**
In general, I found the subject matter of this manuscript interesting and of value to the scientific community. The interpretation of the data and how much it supports the conclusion that "kAE1 variants increases pHi which alters mitochondrial function and leads to reduced cellular energy levels that eventually attenuate energy-dependent autophagic pathways" is largely incomplete. There are significant concerns about the quantification of Western blot data. Additionally, including the R607H variant in the in vitro experiments would improve the interpretation and extrapolation of in vitro data to the kidney.

We apologize for the confusion with R589H and R607H variants. The R607H mutant is the murine ortholog to the human R589H dRTA variation. To clarify this, we have added this information on line 180, in addition to lines 118-119 and line 70.

Suggestions:(1) Can an anion replacement experiment be performed in the mIMCD cells (no Cl or no HCO3) to determine that bicarbonate transport through AE1 is responsible for the reduced ATP rates in Figure 5? Inclusion of WT +dox control would be helpful to convince the reader of the effects.

Because Seahorse real-time cell metabolism ATP rates measurements require specific and patented buffers with un-specified compositions, it was not possible to modify the Cl⁻ or HCO₃⁻ content during the ATP measurement assay. All cell lines, including empty vector cells (EV) were treated with doxycycline; thus, WT + dox was already included. The empty vector cell line treated with doxycycline allowed the exclusion of specific effects of doxycycline on mitochondrial activity as a control. This is now clarified in Figure 5 legend, lines 655-656.

(2) Can the authors measure pHi in fresh kidney sections from the R607H mouse?

Unfortunately, we are not currently able to measure pHi in fresh kidney sections and although we recognize it would benefit greatly to our study, establishing a new collaboration to perform this measurement would significantly delay the publication of this work; therefore, these results will not be available for the present manuscript.

(3) Does pH 7.0 media have any effect on autophagy, as shown in Figure 3? Why was pH 6.6 selected?

The idea was to artificially acidify pHi in mutant cell lines (that have a steady state alkaline pHi) and assess whether this acidification corrects autophagy defects. We first determined that incubation in cell culture medium at pH 6.6 with 0.033 µM nigericin (final potassium concentration: 168 mM) for 2 hours provided optimal conditions, i.e. ensuring cell viability over the 2-hour period while effectively lowering intracellular pH to 6.9, as demonstrated in Supplementary Figure 1A-C.

(4) In vitro experiments should be performed on polarized cells with kAE1 properly inserted in the basolateral membrane. Experiments on subconfluent, non-polarized cells do not support the hypothesis that transport functions of AE1 initiate the cascade of events attributed to these SLC4A1 mutations.

To address this point, we have performed cell surface biotinylations on 70-80 % confluent mIMCD3 cells expressing kAE1 WT, S525F or R589H mutants and show that cell surface abundance of the mutants is not significantly different from the WT protein. This is now shown in Figure 3 A&B. As cell surface biotinylation provides a more quantitative assessment of protein cell surface abundance, we have removed the immunofluorescence images from non-polarised cells and replaced them with representative immunoblots from a cell surface biotinylation assay.

Concerns:(1) No information about the B1 ATPase antibody used.

Now provided in Supplementary Material, ATP6V1B1 Antibody from Bicell cat#20901.

(2) No actin band in Figure 1E (as prepared).

Actin bands are provided for each blot in Figure 1D.

(3) Figures 1E and 1F are labelled wrong in the figure versus the results section.

Thank you for letting us know, this is now corrected.

(4) The cortical sections shown in Figure 4 for the KI/KI do not appear to have the morphology of a CCD. The authors may want to consider including glomeruli to convince the reader of the localization of the tubules. Same concern with Figure 5G and I. The WT image in 5G does not have the morphology of a CCD. Principal cells should be predominant, and ICs should be dispersed.

Both figures 4 and 5 have been updated with images showing glomeruli (light blue “G” on figure) with neighbour and dispersed IC staining.

(5) The quantification of LAMP1 in Figure 4 is unclear. How did the authors determine the boundary of AICs, and how did they calculate the volume of lysosomes? If a zstack was used, how are the authors sure that their 10um section includes the entire AIC?

The quantification of LAMP1 is detailed under “Image analysis”, then “Volocity” sections in Supplementary Material. The boundary of A-IC was manually detected in Volocity based on the presence of the H^+^-ATPase before Volocity analysis for lysosomal volume as described in the Methods.

The 10 micron sections are expected to include full AIC as well as partial AIC, but the frequency of these events should be the same between WT and variants’ sections, therefore they were all included in the analysis if cells displayed H^+^-ATPase signal.

(6) Figure 5: There is no description of how ATP rates are calculated from the provided traces.

We used Agilent Seahorse XF ATP rate assay kit for this experiment. In this assay, the total ATP rate is the sum of ATP production rate from both glycolysis and oxidative phosphorylation. Glycolysis releases protons in a 1:1 ratio with ATP hence the glycolytic ATP rate is calculated from the glycolytic proton efflux rate (glycoPER). GlycoPER is determined by subtracting respiration linked proton efflux from total proton efflux by inhibiting complex I and III. This information is now added to Supplementary Material, in the “Metabolic Flux analysis” section.

(7) Figure labels in Figure 5 are wrong. It seems 5H (as presented) should actually be labeled 5G. In 5H (G?), why did some cells not have any TOM20 pixel intensity for S525F and R589H variants?

Confocal image acquisition in this experiment was kept under the same settings to allow comparison between samples. Therefore, some cells show dimer fluorescence than others. From the figure 5 panels, all cells showed TOM 20 pixel intensity. Figure 5H panel has been relabelled Figure 5G.

(8) In Figure 2, the summary graphs show analysis of more samples than are visible on the included western blots. What is the rationale for this? Why does S525F have 9 samples in BafA1 while R295H only has 3 (2H)? Yet, R295H has 6 samples in 2I. In 2D, S525F has at least 9 samples. Explain.

Figure 2A-C shows representative immunoblots, among several ones independently conducted. Therefore, the final number of samples is higher than showed on Figure 2. This is now indicated in Figure 2 legend, line 603. It became clear quite early in our study that the recessive kAE1 R295H variant does not behave similarly to the other variants studied, maybe because it affects the cytosolic domain, so we did not perform as many replicates for this variant as we did for the others. However, we felt it was valuable to the research community to report the characterization of this variant and decided to keep it in our study.

(9) In general, the actin loading does not appear to be equal between samples. And some figures show the same actin blot twice (2A, C) while some show independent actin bands for LC3B and p62. Equal loading seems a fairly significant control, considering the importance of quantification in the figures.

In addition to performing protein assays, we systematically conduct immunoblot with anti-b-actin antibody to control for loading variability. When possible, two or three proteins, including actin, are detected on the same blot, when molecular weight differ enough. This sometimes results in b-actin being used as a loading control for two different proteins, as seen on Figure 2A and 2C. This is now indicated on lines 605606.

(10) In the Supplemental Figure 2, which band is being quantified for mature CTSD at 33kDa? Same for intermediate CTSD. The quantification of V-ATPase seems questionable based on the actin variance shown in the blot. Surely the ratio of the fourth sample is greater than 1.

Supplementary Figure 2 has been updated to include arrows indicating which band was selected for the quantification. After verifying the measurements of band intensities from “Image Lab” quantification software, we confirm the results, including that fourth KI/KI sample has a ratio of 0.78 (Adj Total Band Vol (Int), lanes 10). Screen shots of quantifications are attached below.

**Author response image 1. sa4fig1:** 

**Author response image 2. sa4fig2:** 

(11) Why are the experiments performed on non-confluent IMCD cells? Figure 1D shows good basolateral localization of AE1, yet the other experiments in the manuscript appear to use IMCD cells in low confluent states, without proper localization of AE1. Figure 3A shows AE1 dispersed throughout the cytoplasm. Why have the authors decided to study the effects of an anion exchanger without it being properly localized to the basolateral membrane? Shouldn't all experiments be performed in polarized IMCDs? If AE1 isnt properly in the membrane, and the cells do not have defined apico-basolateral polarity, then what role can AE1-mediated intracellular pH change have on the results of the experiments? Were the pHi experiments in 3E performed on polarized cells? Or even 1F?

To address this point, we have performed cell surface biotinylations on 70-80 % confluent mIMCD3 cells expressing kAE1 WT, S525F or R589H mutants and show that cell surface abundance of the mutants is not significantly different from the WT protein. This is now shown in Figure 3A & B. As it provides a more quantitative assessment of protein cell surface abundance, we have removed the immunofluorescence images from non-polarised cells and replaced them with a representative immunoblot from a cell surface biotinylation assay.

(12) As mentioned in the public comments, how is the ratio A/(A+B) greater than 1? With A and B > 0. In Figure 3, the data is reasonable, but in Figure 2, the data is simply impossible. What is the explanation for this phenomenon? Why was this presentation of data approved? Is it supposedly a fold of WT, like 2K and 2L? Is the reader also to believe that total LC3B is 2-fold greater in KI/KI mice, as shown in 2K? My eyes, though not densitometry equipment, cannot confirm this. The actin bands are not equal. Yet again, there are 4 lanes of KI/KI mice, but the quantification shows 5 samples.

The ratios in figure 2D, 2F, 2H and 2L have been re-calculated and corrected. As indicated above, immunoblots are representative and quantification of additional blots has been included in the graphs.

(12) Spelling error Figure 4B: cels.

Corrected

References

(1) Mumtaz, R. et al. Intercalated Cell Depletion and Vacuolar H+-ATPase Mistargeting in an Ae1 R607H Knockin Model. Journal of the American Society of Nephrology 28, 1507–1520 (2017).